# LncAABR07053481 inhibits bone marrow mesenchymal stem cell apoptosis and promotes repair following steroid-induced avascular necrosis

Tao Wang[1,2,6], Zhi-Hong Xie[2,6], Lei Wang [1,2], Hong Luo[2], Jian Zhang[1,2], Wen-Tao Dong[1,2], Xiao-Han Zheng[1,2], Chuan Ye[1,2], Xiao-Bin Tian[1,2], Gang Liu[1,2], Xue-Song Zhu[3], Yan-Lin Li[4], Qing-Lin Kang[5], Fei Zhang[1,2✉] & Wu-Xun Peng [1,2✉]

The osteonecrotic area of steroid-induced avascular necrosis of the femoral head (SANFH) is a hypoxic microenvironment that leads to apoptosis of transplanted bone marrow mesenchymal stem cells (BMSCs). However, the underlying mechanism remains unclear. Here, we explore the mechanism of hypoxic-induced apoptosis of BMSCs, and use the mechanism to improve the transplantation efficacy of BMSCs. Our results show that the long non-coding RNA AABR07053481 (LncAABR07053481) is downregulated in BMSCs and closely related to the degree of hypoxia. Overexpression of LncAABR07053481 could increase the survival rate of BMSCs. Further exploration of the downstream target gene indicates that LncAABR07053481 acts as a molecular "sponge" of miR-664-2-5p to relieve the silencing effect of miR-664-2-5p on the target gene *Notch1*. Importantly, the survival rate of BMSCs overexpressing LncAABR07053481 is significantly improved after transplantation, and the repair effect of BMSCs in the osteonecrotic area is also improved. This study reveal the mechanism by which LncAABR07053481 inhibits hypoxia-induced apoptosis of BMSCs by regulating the miR-664-2-5p/*Notch1* pathway and its therapeutic effect on SANFH.

[1] Department of Orthopedics and Traumatology, The Affliated Hospital of Guizhou Medical University, Guiyang, Guizhou 550004, P.R. China. [2] School of Clinical Medicine, Guizhou Medical University, Guiyang, Guizhou 550004, P.R. China. [3] Department of Orthopedic Surgery, The First Affiliated Hospital of Soochow University, Suzhou, Jiangsu 215000, P.R. China. [4] Department of Sports Medicine, First Affiliated Hospital of Kunming Medical University, Kunming, Yunnan 650000, P.R. China. [5] Department of Orthopedics, Shanghai Jiao Tong University Affiliated Sixth People's Hospital, Shanghai 200233, P.R. China. [6] These authors contributed equally: Tao Wang, Zhi-Hong Xie. ✉email: 1426287582@qq.com; pwx2021163@163.com

Steroid-induced avascular necrosis of the femoral head (SANFH) is a common orthopedic complication after high-dose steroid hormone treatments[1,2]. SANFH ranks first among all types of non-traumatic femoral head necrosis and is one of the main diseases endangering human health worldwide[3,4]. Furthermore, 80% of untreated patients are prone to collapse of the femoral head structure and subchondral fractures in the late stage, and the disability rate is high[5–7]. Therefore, early prevention and treatment of SANFH are particularly important to reduce hip disability. Preclinical studies have shown that bone marrow mesenchymal stem cell (BMSC) transplantation may become an ideal treatment for early SANFH[8–11]. However, the transplanted BMSCs display high levels of apoptosis in the osteonecrotic area in vivo, which limits the transplantation efficacy[12,13]. The apoptosis of transplanted BMSCs is mainly related to the local hypoxic microenvironment in the necrotic area of the femoral head[14–17]. There is still a lack of effective intervention methods for hypoxia-induced apoptosis of transplanted BMSCs. How to effectively inhibit hypoxia-induced apoptosis of transplanted BMSCs and improve the efficacy of BMSC transplantation is a challenge for further research to overcome.

In the research of diseases related to ischemia-hypoxia, many non-coding RNAs (ncRNAs) have been found to play key roles in regulating hypoxia-induced apoptosis[18–20]. Long non-coding RNA (LncRNAs) belongs to a class of ncRNAs with transcripts that are between 200 nt and 100 kb in length. Many studies have shown that LncRNAs are widely involved in various biological processes, such as chromosome silencing, chromatin modification, transcriptional activation, transcriptional interference, and intranuclear transport[21–23]. LncRNA expression levels are regulated by parameters of the cell growth environment, such as hypoxic conditions. For example, under hypoxic conditions, LncEFNA3 can promote the proliferation of breast cancer cells from blood vessels to surrounding tissues[24]. Moreover, LncHOTTIP promotes epithelial-mesenchymal transformation under hypoxia[25], and LncMALAT1 enhances arsenic-induced glycolysis under hypoxia[26]. Furthermore, the differential expression of LncRNAs may be closely related to cell function under hypoxic conditions. However, it is unclear whether there are differentially expressed LncRNAs that regulate hypoxia-induced apoptosis of BMSCs transplanted to the necrotic area of the femoral head. Therefore, searching for LncRNAs related to hypoxia-induced apoptosis of BMSCs and understanding their biological functions and regulatory mechanisms will help reveal new anti-apoptotic targets and provide a novel method to reduce hypoxia-induced apoptosis of transplanted BMSCs.

In this study, we analyzed the expression profile of LncRNAs in BMSCs using microarray data and found that LncAABR07053481 was associated with hypoxia-induced apoptosis of BMSCs. The role and mechanism of LncAABR07053481 as a competing endogenous RNA (ceRNA) of mir-664-2-5p, and its effect on the *Notch1* pathway and the inhibition of hypoxia-induced apoptosis of BMSCs were determined, and the repair efficacy of inhibiting hypoxia-induced apoptosis of BMSCs in early SANFH was evaluated. Our findings provide a novel idea and method for improving the efficacy of BMSC transplantation.

## Results

### LncAABR07053481 is downregulated under hypoxia and associated with apoptosis of BMSCs.

To study the effect of hypoxia on BMSCs, we successfully isolated rat BMSCs (Supplementary Fig. 1) and constructed an early SANFH rat model (Supplementary Fig. 2a, b). Consistent with previous studies[27,28], our data showed that the oxygen concentration in the necrotic area

of the femoral head was <0.1% (Supplementary Fig. 2c). Subsequently, BMSCs were cultured under hypoxia (0% $O_2$) for 48 h to simulate a hypoxic microenvironment, and we found that, in the hypoxic environment, the content of reactive oxygen species (ROS) in BMSCs was increased (Supplementary Fig. 2d, e), the mitochondrial membrane potential was decreased (Supplementary Fig. 2f, g), ATP synthesis was decreased (Supplementary Fig. 2h), the expression of apoptosis-related proteins was increased (Supplementary Fig. 2i–l), and there were a large number of apoptotic BMSCs (Supplementary Fig. 2m–o). This indicates that hypoxia can lead to the apoptosis of BMSCs. Studies have found that LncRNA plays an important role in the regulation of hypoxia-induced apoptosis[29,30]. Therefore, to reveal the LncRNAs that regulate BMSC apoptosis under hypoxic conditions, BMSCs were cultured under normoxia (20% $O_2$) or hypoxia (0% $O_2$) conditions for 48 h, and the LncRNA profiles of BMSCs exposed to normoxia or hypoxia were obtained by microarray analysis. Our results showed that 106 LncRNAs were upregulated and 173 LncRNAs were downregulated under hypoxic conditions, with 33 potential LncRNAs, in particular, being upregulated or downregulated by >3-fold after hypoxia, as shown in the heatmap (Fig. 1a). We focused on a LncRNA transcript that was highly downregulated during hypoxia (i.e., LncAABR07053481, transcript ID: ENSRNOT00000078988.1, source: Ensembl; the LncRNA transcript also matches LOC120101634, Accession: XR_005502377, source: RefSeq). LncAABR07053481, located on chromosome 3, is composed of six exons with a full-length 2112-nt transcript. Of note, LncAABR07053481 lacks the ability to encode protein (Fig. 1b). In addition, it does not produce any detectable peptides according to the prediction made with CPAT and by in vitro translation assay (Fig. 1c, d). Subsequently, to verify the microarray data, we detected the expression of LncAABR07053481 in the BMSC hypoxia model by qPCR, and the results were consistent with the microarray data. Specifically, we found that the expression of LncAABR07053481 was significantly downregulated under hypoxia, and, with the aggravation of hypoxia, the apoptosis rate of BMSCs increased, while the expression of LncAABR07053481 was continuously downregulated (Fig. 1e–g). In vitro, our results showed that the expression of LncAABR07053481 in BMSCs was downregulated under hypoxia, suggesting that LncAABR07053481 may be closely related to hypoxia-induced apoptosis of BMSCs. We also verified the expression of LncAABR07053481 in vivo and the relationship between LncAABR07053481 and the apoptosis of transplanted BMSCs. In vivo, we found that, compared with the control (Normal) group, the expression of LncAABR07053481 in the BMSCs transplanted with the model (SANFH) group was significantly downregulated (Fig. 1h). Moreover, the expression of LncAABR07053481 in the BMSC transplantation group was continuously downregulated with the prolongation of transplantation time in vivo (Fig. 1i). Thus, both in vitro and in vivo data showed that the differential expression of LncAABR07053481 may be closely related to hypoxia-induced apoptosis of BMSCs.

### LncAABR07053481 can inhibit hypoxia-induced apoptosis of BMSCs.

To further clarify the role of LncAABR07053481 differential expression in the apoptosis of BMSCs, we transfected BMSCs with LncAABR07053481-overexpression lentivirus (Lv-LncAABR07053481) and LncAABR07053481-knockdown lentivirus (shLV-LncAABR07053481), respectively. The results of qPCR showed that LncAABR07053481 was successfully overexpressed or knocked down in the BMSCs (Fig. 2a). Then, to verify the key role of LncAABR07053481 in the regulation of apoptosis in BMSCs, we first knocked down LncAABR07053481 in BMSCs under normoxia, and the results showed that, under normoxia, the LncAABR07053481-knockdown group (sh AABR07053481) exhibited increased BMSC

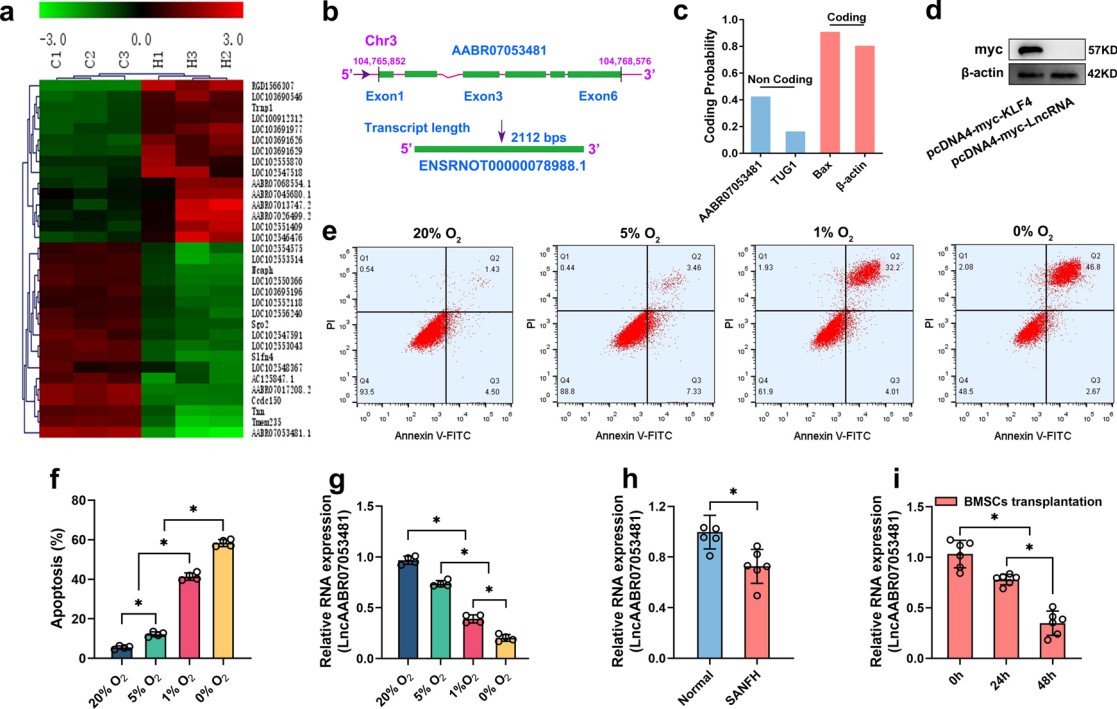

**Fig. 1 LncAABR07053481 is downregulated under hypoxia and associated with apoptosis of BMSCs. a** Cluster analysis of LncRNAs ($n = 3$). **b** Location of LncRNA in the genome. **c** Prediction of LncAABR07053481 coding capability by the coding-potential assessment tool (CPAT). **d** The expression of the Myc-fused protein was analyzed by immunoblotting with an anti-Myc antibody ($n = 3$). **e, f** The relationship between the degree of hypoxia and apoptosis ($n = 4$). **g** Expression of LncAABR07053481 under different oxygen concentrations ($n = 4$). **h** Detection of LncAABR07053481 expression in the normal femoral head and the area of osteonecrosis by qPCR ($n = 6$). **i** Detection of LncAABR07053481 expression in BMSCs at different times after transplantation by qPCR ($n = 6$). Data were shown as mean ± SD. $^*P < 0.05$; In (**h**), statistical significance was calculated by Student's $t$-test; in (**f, g, i**), statistical significance was calculated by one-way ANOVA with Tukey's post hoc tests.

apoptosis compared with the empty vector group (Vec) (Fig. 2b, c). We next subjected BMSCs to hypoxia (0% $O_2$) for 48 h. Notably, compared with the empty vector group, overexpression of LncAABR07053481 in BMSCs (Oe AABR07053481) significantly reduced the hypoxia-induced apoptosis of BMSCs (Fig. 2d, e). Consistently, the protein level of caspase-3 (CASP-3) was also decreased, while the expression levels of the anti-apoptosis–related proteins Survivin and B-cell lymphoma 2 (Bcl-2) were increased in BMSCs overexpressing LncAABR07053481 (Fig. 2f–i). Conversely, knockdown of LncAABR07053481 in BMSCs (sh AABR07053481) significantly increased the level of CASP-3 protein, inhibited the expression of Survivin and Bcl-2, and further aggravated the hypoxia-induced apoptosis of BMSCs (Fig. 2d–i).

To further determine the anti-apoptosis effect of LncAABR07053481 in vivo, we overexpressed or knocked down LncAABR07053481 in BMSCs, then labeled BMSCs with 1,1-dioctadecyl-3,3,3,3-tetramethylindotricarbocyanine iodide (DiR) fluorescence to track the viable BMSCs in vivo (i.e., positive DiR-labeled cells were viable). Finally, BMSCs were transplanted to treat the SANFH model rats, and the survival of BMSCs in vivo was detected using the Spectrum IVIS. At 48 h after BMSC transplantation, we found that the DiR fluorescence intensity in the operative area of the overexpression group (LD/Oe AABR07053481) was significantly higher than that in the empty vector group (LD/Vec), while that in the knockdown group (LD/sh AABR07053481) was significantly decreased (Fig. 2j, k). Furthermore, the results of TUNNEL staining showed that the LncAABR07053481-overexpression group displayed a significantly reduced apoptosis rate of transplanted BMSCs, while LncAABR07053481 knockdown increased the apoptosis of transplanted BMSCs in the transplantation area (Fig. 2l, m).

Together, these findings suggest that LncAABR07053481 can inhibit hypoxia-induced apoptosis of transplanted BMSCs in vitro and in vivo.

**LncAABR07053481 inhibits hypoxia-induced apoptosis of BMSCs by regulating the *Notch1* pathway.** To establish the mechanism by which LncAABR07053481 inhibits hypoxia-induced apoptosis of BMSC, we performed microarray analysis on BMSCs overexpressing LncAABR07053481. The results showed that the gene-expression profile differed between the LncAABR07053481-overexpression group and the empty virus group (Fig. 3a). KEGG pathway analysis and GSEA demonstrated that these differentially expressed genes were significantly enriched in the *Notch1* pathway. Moreover, the *Notch1* pathway had the highest enrichment scores, in which *Notch1* mRNA was significantly upregulated after LncAABR07053481 overexpression (Fig. 3b, c). We next verified the microarray data using qPCR and western blot analysis, and the results showed that the expression of *Notch1* and *Notch1* intracellular domain (NICD1) in BMSCs overexpressing LncAABR07053481 were significantly upregulated under hypoxia (Fig. 3d–g). Previous studies have shown that the activation of the *Notch1* signaling triggers the cleavage of *Notch1* receptors, causing NICD1 (the intracellular active fragment of *Notch1*) to release from the cell membrane and into the nucleus, which can inhibit apoptosis by upregulating the transcription and expression of *Notch1* target genes (Survivin and Bcl-2)[31–34]. Therefore, these data suggest that LncAABR07053481 promotes the expression of anti-apoptotic proteins such as Survivin and Bcl-2 by activating the *Notch1* pathway and finally inhibits the hypoxia-induced apoptosis of BMSCs.

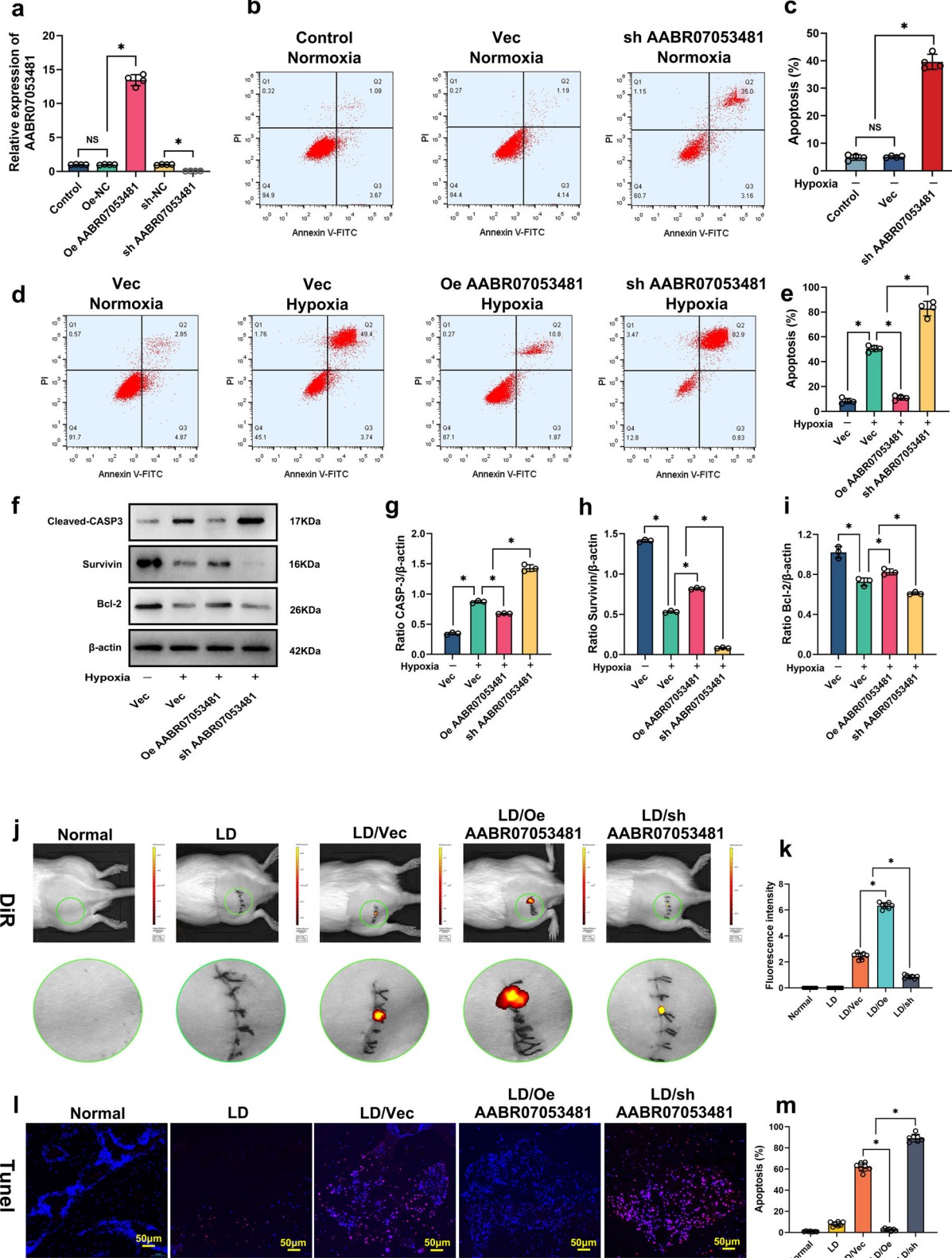

**Fig. 2 LncAABR07053481 can inhibit hypoxia-induced apoptosis of BMSCs. a** The expression of LncAABR07053481 was detected by qPCR ($n = 4$). **b–e** Detection of BMSC apoptosis under different conditions by Annexin V-fluorescein isothiocyanate (FITC, green)/propidium iodide (PI, red) ($n = 4$). **f–i** The expression levels of CASP-3, Survivin, and Bcl-2 were detected by western blotting ($n = 3$). **j** Forty-eight hours after surgery, the fluorescence intensity of 1,1-dioctadecyl-3,3,3,3-tetramethylindotricarbocyanine iodide (DiR, red) in the transplantation area was detected by live imaging of small animals ($n = 7$). **k** Quantitative analysis of DiR fluorescence intensity in the transplanted area ($n = 7$). **l–m** At 48 h after surgery, TdT-mediated dUTP nick-end labeling (TUNEL, red) /4′,6-diamidino-2-phenylindole (DAPI, blue) was used to detect apoptosis in the transplantation area ($n = 7$). Data were shown as mean ± S.D. $^*P < 0.05$; in (**a, c, e, g–i, k, m**), statistical significance was calculated by one-way ANOVA with Tukey's post hoc test.

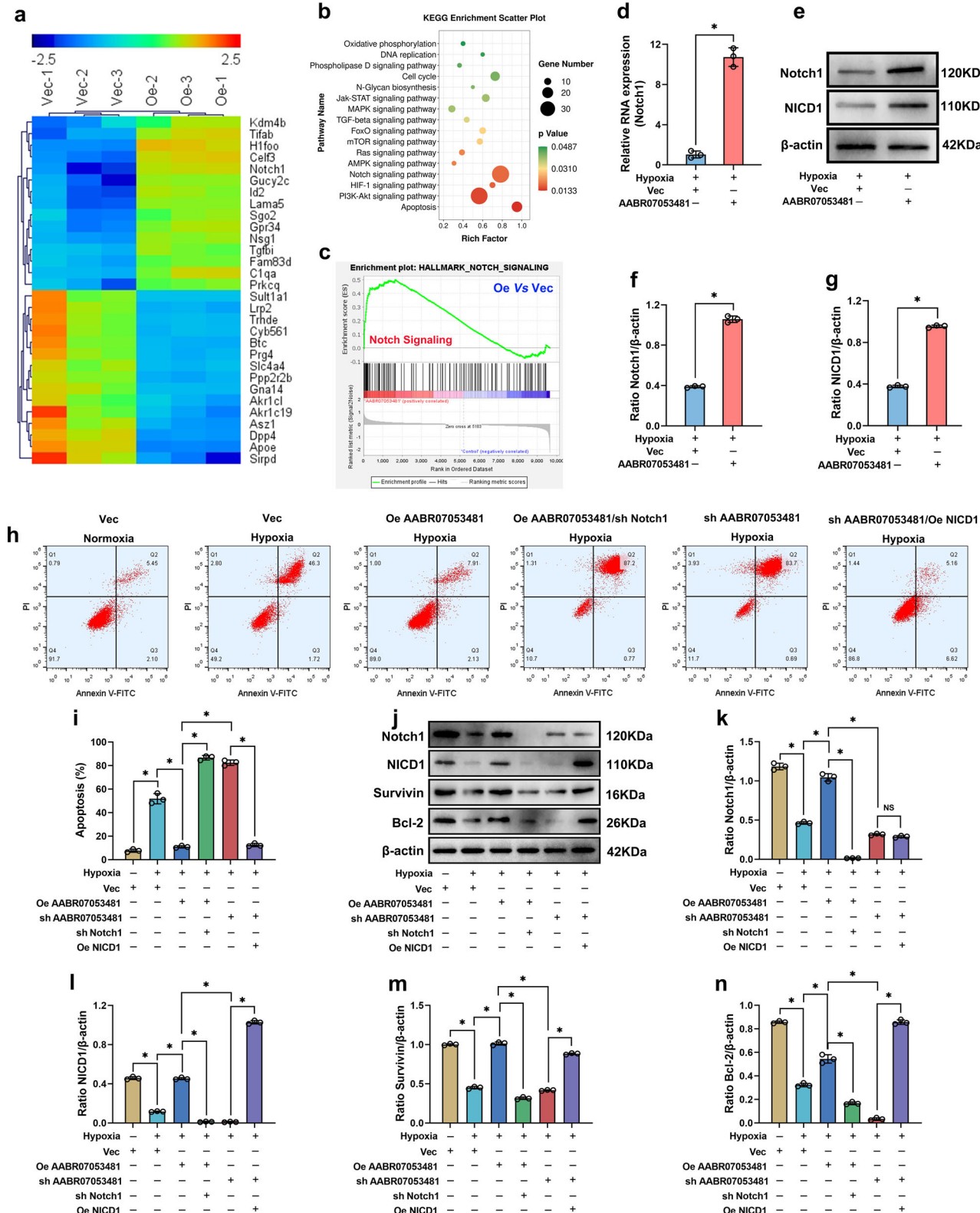

**Fig. 3 LncAABR07053481 inhibits hypoxia-induced apoptosis of BMSCs by regulating the *Notch1* pathway. a** mRNA clustering analysis (*n* = 3). **b** Kyoto Encyclopedia of Genes and Genomes (KEGG) pathway enrichment analysis of differentially expressed genes. **c** Gene set enrichment analysis (GSEA). **d** The expression of *Notch1* was detected by qPCR (*n* = 3). **e–g** The expression levels of *Notch1* and NICD1 were detected by western blotting (*n* = 3). **h, i** Detection of BMSC apoptosis by Annexin V-FITC/PI (*n* = 3). **j–n** The expression levels of *Notch1*, NICD1, Survivin, and Bcl-2 were detected by western blotting (*n* = 3). Data were shown as mean ± SD. *P < 0.05; In (**d**, **f**, **g**), statistical significance was calculated by Student's *t*-test; In (**i**, **k–n**), statistical significance was calculated by one-way ANOVA with Tukey's post hoc test.

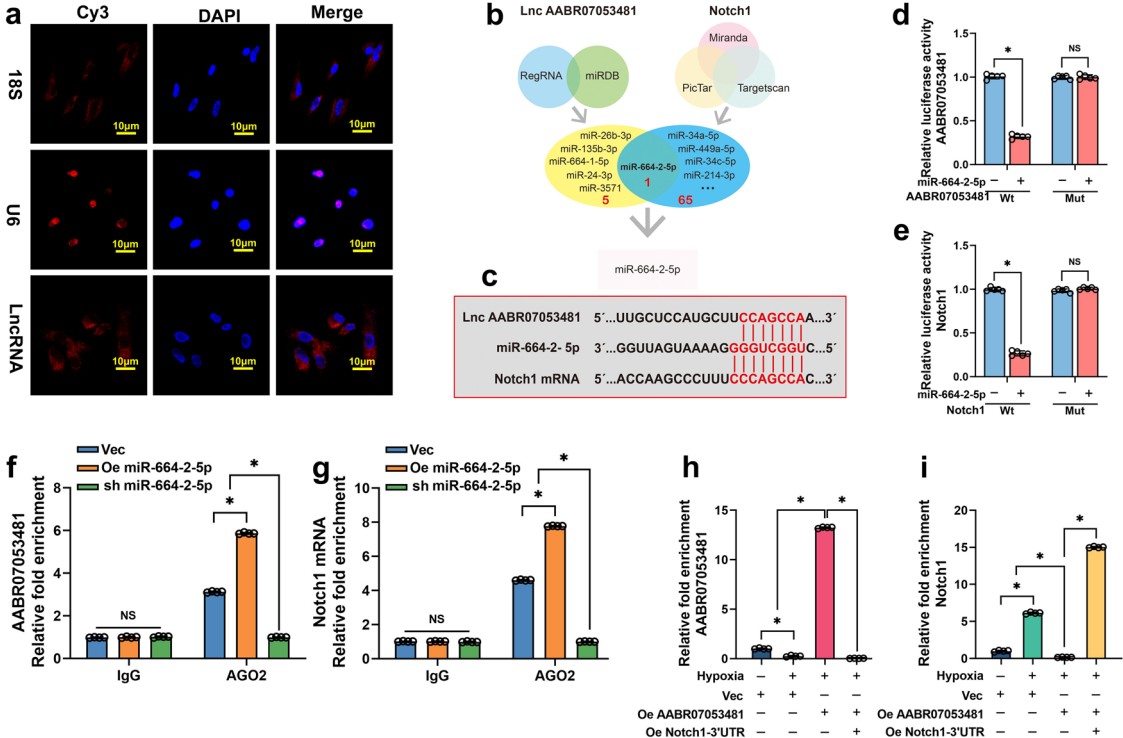

**Fig. 4 LncAABR07053481 functions as a ceRNA and sponges miR-664-2-5p in BMSCs. a** RNA fluorescence in situ hybridization (RNA-FISH) detection of the subcellular localization of LncAABR07053481, 18S, and U6 were used as positive control. (n = 4). **b** Bioinformatics tools predict the miRNAs that may bind to the untranslated region at the 3′ end of *Notch1* mRNA (*Notch1* mRNA 3′UTR) and LncAABR07053481. **c** Binding sites of miR-664-2-5p with LncAABR07053481 and *Notch1* mRNA 3′UTR. **d, e** Luciferase reporter assay shows that miR-664-2-5p binds to LncAABR07053481 and *Notch1* mRNA (n = 5). **f–h** RNA immunoprecipitation (RIP) detection of LncAABR07053481 enrichment to miRNA ribonucleoprotein complexes (miRNPs) (n = 4). **g–i** RIP detection of enrichment of *Notch1* mRNA to miRNPs (n = 4). Data were shown as mean ± SD. *P < 0.05; In (**d–i**), statistical significance was calculated by one-way ANOVA with Tukey's post hoc test.

Subsequently, we verified the role of the *Notch1* pathway in the regulation of hypoxia-induced apoptosis of BMSCs. The results showed that under hypoxia, overexpression of LncAABR07053481 in BMSCs significantly increased the levels of *Notch1* and NICD1, upregulated the expression of Survivin and Bcl-2, and reduced the apoptosis of BMSCs (Fig. 3h–n). Knockdown of the *Notch1* gene in BMSCs overexpressing LncAABR07053481 blocked the anti-apoptosis of LncAABR07053481, downregulated the expression of Survivin and Bcl-2, and increased the apoptosis of BMSC (Fig. 3h–n). By contrast, in the overexpression of NICD1 on the basis of LncAABR07053481 knockdown to activate the *Notch1* pathway, hypoxia-induced apoptosis of BMSCs was significantly inhibited, and the expression levels of Survivin and Bcl-2 were significantly upregulated (Fig. 3h–n). These findings suggest that LncAABR07053481 can inhibit hypoxia-induced apoptosis of BMSCs by regulating the *Notch1* pathway.

**LncAABR07053481 functions as a ceRNA and sponges miR-664-2-5p in BMSCs.** To investigate the molecular mechanism by which LncAABR07053481 regulates the *Notch1* pathway, we first confirmed that LncAABR07053481 was mainly located in the cytoplasm by RNA-FISH (Fig. 4a). Increasing evidence has shown that classic regulation of LncRNA in the cytoplasm occurs as a result of ceRNAs; this removes the silencing effect of miRNA on target genes by adsorbing miRNA and finally promotes the expression of target genes[35,36]. As a result, we considered whether LncAABR07053481 regulates and activates *Notch1* through similar mechanisms. To test this hypothesis, we used bioinformatics tools, such as Targetscan and RegRNA, to predict the possible

binding miRNAs of the untranslated region at the 3' end of *Notch1* mRNA (*Notch1* mRNA 3'UTR) and LncAABR07053481. The results show that miR-664-2-5p is a potential target for the combination of LncAABR07053481 and *Notch1* mRNA 3'UTR (Fig. 4b). In addition, prediction by bioinformatics tools demonstrated that LncAABR07053481 and *Notch1* mRNA 3′UTR share a consensus binding site on miR-664-2-5p (Fig. 4c). Therefore, we speculated that LncAABR07053481 acts as the ceRNA of miR-664-2-5p.

Next, we applied a luciferase assay to detect the specific binding sites between the LncAABR07053481/*Notch1* mRNA 3′UTR and miR-664-2-5p. We mutated miR-664-2-5p putative binding sites in LncAABR07053481, and then constructed dual-luciferase reporter vectors that contained the LncAABR07053481 mutant (LncAABR07053481-Mut) or wild-type (LncAABR07053481-Wt). Figure 4d shows that the relative luciferase activity of LncAABR07053481-Wt in BMSCs was obviously reduced after co-transfection of miR-664-2-5p, but did not change the activity of the mutant vector, suggesting that miR-664-2-5p is a direct target of LncAABR07053481. Similarly, we constructed a luciferase reporter vector that contained wild-type or mutant miR-664-2-5p putative binding sites in the *Notch1* mRNA 3′UTR (*Notch1* mRNA 3′UTR-Wt and *Notch1* mRNA 3′UTR-Mut). The alignment of the complementary binding of miR-664-2-5p and the 3′UTR of *Notch1* was confirmed by the luciferase assay (Fig. 4e). These results confirmed that miR-664-2-5p could specifically bind to LncAABR07053481 and the *Notch1* mRNA 3′ UTR at the predicted binding sites.

To confirm the competitive relationship between LncAABR07053481 and *Notch1* mRNA, RIP experiments were

conducted to detect the enrichment of LncAABR07053481 and *Notch1* mRNA to miRNA ribonucleoprotein complexes (miRNPs) in BMSC extracts using an AGO2 antibody. The results showed that both LncAABR07053481 and *Notch1* mRNA were specifically enriched in AGO2 antibody-associated complexes, but not in control IgG (Fig. 4f, g). Moreover, the enrichment of LncAABR07053481 and *Notch1* mRNA to miRNPs increased after the overexpression of miR-664-2-5p, while the enrichment of LncAABR07053481 and *Notch1* mRNA decreased following the knockdown of miR-664-2-5p (Fig. 4f, g). We also found that the overexpression of LncAABR07053481 in BMSCs significantly increased the enrichment of LncAABR07053481 to miRNPs (Fig. 4h), but significantly decreased the enrichment of *Notch1* mRNA to miRNPs (Fig. 4i). By contrast, overexpression of the 3′UTR of *Notch1* mRNA increased the enrichment of *Notch1* mRNA to miRNPs (Fig. 4i), while the enrichment of LncAABR07053481 to miRNPs was decreased significantly (Fig. 4h). Taken together, these data show that LncAABR07053481 functions as a ceRNA for miR-664-2-5p in the regulation of hypoxia-induced BMSC apoptosis.

**The LncAABR07053481/miR-664-2-5p/*Notch1* axis is involved in the regulation of hypoxia-induced apoptosis of BMSCs.** To further verify the regulatory role of the LncAABR07053481/miR-664-2-5p/*Notch1* axis in hypoxia-induced apoptosis of BMSCs, we overexpressed LncAABR07053481 in BMSCs (Fig. 5a), then subjected the cells to hypoxia for 48 h. The results showed that, compared with the empty vector group, the upregulation of LncAABR07053481 significantly increased the expression of both *Notch1* and NICD1 (Fig. 5d–f) and reduced the hypoxia-induced apoptosis of BMSCs (Fig. 5g, h). Next, we upregulated the expression of miR-664-2-5p again in the BMSCs that overexpressed LncAABR07053481 (Fig. 5b). The results showed that, under hypoxic conditions, the expression levels of *Notch1* and NICD1 in BMSCs were significantly downregulated (Fig. 5d–f), while the apoptosis of BMSCs was increased (Fig. 5g, h). This trend may be because the upregulation of miR-664-2-5p increased the enrichment of LncAABR07053481 and *Notch1* to miRNPs, strengthened the silencing of *Notch1* mRNA by miR-664-2-5p, and significantly inhibited the anti-hypoxia-induced apoptosis of LncAABR07053481. Subsequently, we continued to upregulate the expression of NICD1 in BMSCs that overexpressed both LncAABR07053481 and miR-664-2-5p to activate the *Notch1* pathway (Fig. 5c–f). As a result, we found that the apoptosis of BMSCs was decreased significantly following the upregulation of NICD1 (Fig. 5g, h). These results reveal that the upregulation of NICD1 promotes the survival of BMSCs under hypoxic conditions. Overall, our data suggest that the LncAABR07053481/miR-664-2-5p/*Notch1* axis regulates the hypoxia-induced apoptosis of BMSCs.

**LncAABR07053481-overexpressed BMSCs promote the repair of early SANFH.** Determining novel anti-apoptotic targets and inhibiting hypoxia-induced apoptosis of BMSCs will help to improve the repair effect of transplanted BMSCs in early SANFH. Our data indicate that LncAABR07053481 inhibits hypoxia-induced apoptosis of BMSCs in vitro and in vivo. However, it is unclear whether the inhibition of the hypoxia-induced apoptosis of BMSCs by LncAABR07053481 can improve the repair effect in early SANFH. Therefore, we transplanted BMSCs with over-expression or knockdown of LncAABR07053481 to repair early SANFH in a rat model, then further evaluated the efficacy of LncAABR07053481-modified BMSCs on early SANFH. At 12 weeks after BMSC transplantation, we evaluated the repair of the osteonecrotic area by micro-CT examination. Ultimately, the

results showed that, compared with the Normal group, there were different degrees of bone defects in the lesion debridement group (LD), empty vector group (LD/Vec), and knockdown group (LD/sh AABR07053481), and the defect area had not been completely repaired. In the LncAABR07053481-overexpression group (LD/Oe AABR07053481), the bone defect area had been completely filled with bone tissue, and the defect area had been completely repaired, with no significant difference compared with the normal group (Fig. 6a). Similarly, compared with the normal group, the number of trabeculae, the trabecular thickness, and the new bone volume fraction in the osteonecrotic area of the LD group, LD/Vec group, and LD/sh AABR07053481 group were significantly decreased. By contrast, compared with the LD group, LD/Vec group, and LD/sh AABR07053481 group, the LncAABR07053481-overexpression group demonstrated a significantly increased number of trabeculae, trabecular thickness, and new bone volume fraction in the osteonecrotic area (Fig. 6b–d). In addition, the results of H&E and Masson's staining were consistent with those of the micro-CT examination. In the LncAABR07053481-overexpression group, the osteonecrotic area was completely repaired, and the bone tissue tended to mature (Fig. 6e). Immunohistochemical detection of osteogenic markers showed that, compared with the Normal group, the levels of runt-related transcription factor 2 (Runx2) and osteopontin (OPN) were decreased in the LD group, LD/Vec group, and LD/sh AABR07053481 group. The LncAABR07053481-overexpression group also demonstrated significantly increased levels of Runx2 and OPN in the osteonecrotic area compared with those of the LD group, LD/Vec group, and LD/sh AABR07053481 group (Fig. 6f–h). These results confirm that LncAABR07053481 improves the efficacy of transplanted BMSCs in early SANFH.

## Discussion

BMSC transplantation to repair osteonecrosis or bone defect is a new method in the field of orthopedic research, and it brings new hope to patients with disabilities such as SANFH[37–39]. However, the osteonecrotic area is a hypoxic environment, which brings many challenges to the survival of BMSCs and is also an important obstacle to be solved in BMSC transplantation for osteonecrosis[12,27,28,40]. It has been found that various LncRNAs are abnormally expressed in cells in hypoxic conditions, and play important roles in the regulation of cell function[29,30]. In this study, we found that the expression of LncAABR07053481 was significantly downregulated in BMSCs under hypoxia, which was closely associated with the hypoxia-induced apoptosis of BMSCs. Importantly, when we upregulated LncAABR07053481, it was confirmed that LncAABR07053481 inhibited hypoxia-induced apoptosis of BMSCs both in vitro and in vivo. Moreover, the upregulation of LncAABR07053481 led to an elevated expression of *Notch1* by competitively binding to miR-664-2-5p. *Notch1* is an important signaling pathway, which affects cell fate and plays important roles in regulating biological processes such as apoptosis[31–34,41,42]. Our findings not only suggest an important role of LncAABR07053481 acting as an miRNA sponge during the regulation of hypoxia-induced BMSC apoptosis but also highlight the therapeutic potential of BMSC transplantation with overexpression of LncAABR07053481 in the repair of early SANFH.

The human genome is pervasively transcribed, but >80% of RNA transcripts are not translated into proteins, and these are collectively referred to as non-coding genes, of which LncRNAs are the main constituent element[43,44]. These LncRNAs are not only diverse but also complex in function, with roles in many biological processes, including apoptosis, proliferation, differentiation, and metabolism[45–47]. For example, LncMT1DP reduces

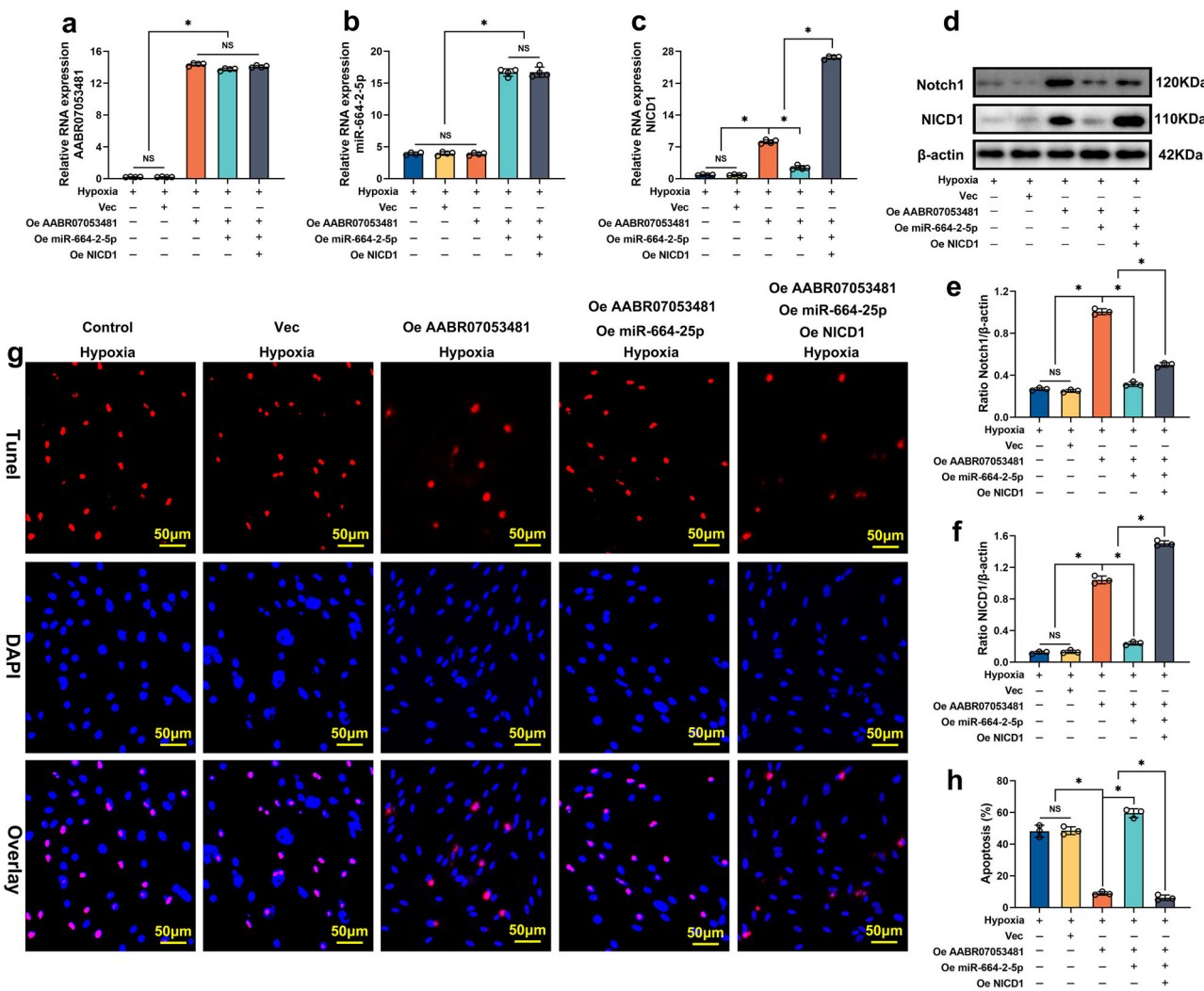

**Fig. 5 The LncAABR07053481/miR-664-2-5p/*Notch1* axis is involved in the regulation of hypoxia-induced apoptosis of BMSCs. a–c** The expressions of LncAABR07053481, miR-664-2-5p, and NICD1 were detected by qPCR ($n = 4$). **d–f** The expression levels of *Notch1* and NICD1 were detected by western blotting ($n = 3$). **g, h** Detection of BMSC apoptosis under different conditions by TUNEL/DAPI ($n = 3$). Data were shown as mean ± SD. *$P < 0.05$; In (**a–c, e, f, h**), statistical significance was calculated by one-way ANOVA with Tukey's post hoc test.

Nrf2 by stabilizing miR-365, a negative regulator of Nrf2, and plays a key role in the Cd-induced apoptosis of hepatocytes[48]. Moreover, SNHG12 acts as a molecular sponge of microRNA to promote tumor proliferation, metastasis, and epithelial-mesenchymal transition[49]. In addition, lncMAR1 has been shown to act as a miR-487b sponge to regulate Wnt5a protein, resulting in muscle differentiation and regeneration[50]. Similarly, in this study, we discovered a key LncRNA in the BMSC hypoxia model, LncAABR07053481, through microarray analysis. Further study showed that the expression of LncAABR07053481 was significantly downregulated in hypoxia compared with normal BMSCs; moreover, its expression was continuously down-regulated with the aggravation of hypoxia and the concomitantly-increased apoptotic rate. As such, we hypothesized that LncAABR07053481 plays a key regulatory role in the hypoxia-induced apoptosis of BMSCs. Functional studies revealed that overexpression of LncAABR07053481 significantly improved the survival rate of BMSCs under hypoxia and inhibited hypoxia-induced apoptosis of BMSCs both in vitro and in vivo.

We next performed a microarray analysis to further study the downstream regulation mechanism of LncAABR07053481. The results showed that the *Notch1* pathway was significantly

activated after LncAABR07053481 overexpression and that LncAABR07053481 was positively correlated with the expression of *Notch1* and its target genes Survivin and Bcl-2. Furthermore, through functional experiments, we verified that LncAABR07053481 could inhibit hypoxia-induced apoptosis of BMSCs by regulating the expression of *Notch1*. As typical non-coding genes, LncRNAs have complex and diverse biological functions. An increasing number of studies have argued that the different biological functions of LncRNAs largely depend on their distinct subcellular localization[51,52]. Accumulated evidence has also shown that LncRNAs located in the cytoplasm can partici-pate in gene regulation at the posttranscriptional level, including by acting as ceRNAs and protecting the target mRNAs from repression[35,53,54]. We used RNA-FISH assays to show that LncAABR07053481 was preferentially localized in the cytoplasm, indicating its potential function as an miRNA sponge. Bioinfor-matics analysis indicated that miR-664-2-5p is a target of LncAABR07053481 and the *Notch1* mRNA 3′UTR. Expectedly, the results of the luciferase assay confirmed that miR-664-2-5p could specifically bind to LncAABR07053481 and the *Notch1* mRNA 3′UTR at the predicted binding sites. We further con-firmed the competitive relationship between LncAABR07053481

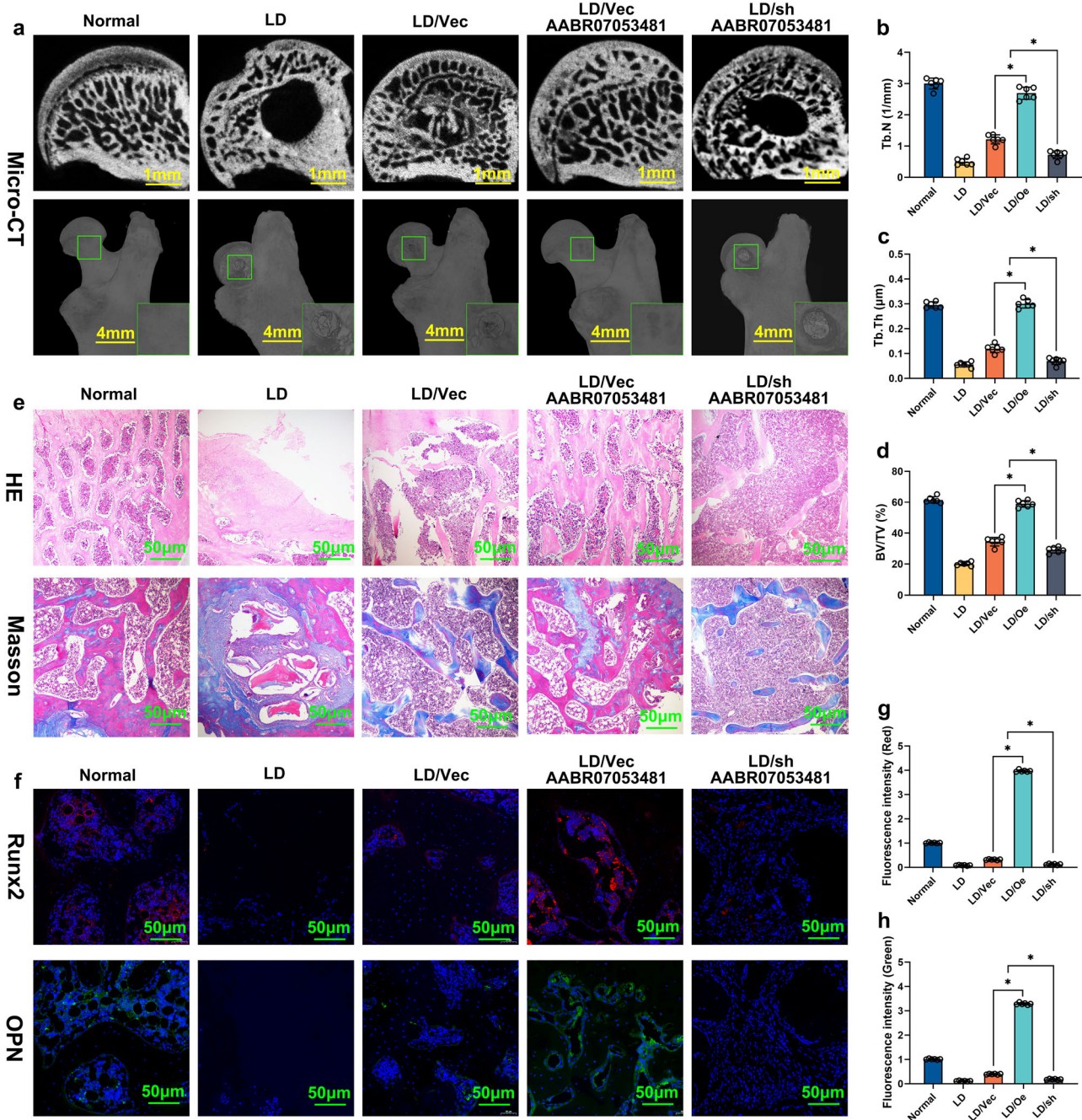

**Fig. 6 LncAABR07053481-overexpressed BMSCs promote the repair of early SANFH. a** At 12 weeks after BMSC transplantation, micro-CT was used to analyze the repair of the necrotic area of the femoral head (*n* = 6). **b-d** Quantitative analysis of the number of trabeculae, the trabecular thickness, and the volume fraction of new bone tissue (*n* = 6). **e** At 12 weeks after BMSC transplantation, H&E and Masson staining were used to evaluate the repair of the necrotic area of the femoral head (*n* = 6). **f** At 12 weeks after BMSC transplantation, immunohistochemistry was used to detect osteogenic markers of runt-related transcription factor 2 (Runx2, Cy3 label, red) and osteopontin (OPN, FITC label, green) (*n* = 6). **g** Quantitative analysis of Runx2 expression (*n* = 6). **h** Quantitative analysis of OPN expression (*n* = 6). Data were shown as mean ± SD. *$P < 0.05$; In (**b-d**, **g**, **h**), statistical significance was calculated by one-way ANOVA with Tukey's post hoc test.

and *Notch1* mRNA by RIP assay. Our results also showed that the overexpression of LncAABR07053481 inhibited hypoxia-induced apoptosis of BMSCs in vitro by upregulating *Notch1* as a ceRNA to sponge miR-664-2-5p. Collectively, our results indicate that a strategy targeting LncAABR07053481 may serve as a valid therapeutic regimen to inhibit hypoxia-induced apoptosis of BMSCs.

SANFH is a complex pathological process in which various factors lead to intramedullary microvascular lesions, affecting the blood and oxygen supply of the femoral head and ultimately

causing hypoxia in the necrotic area[14–17]. The hypoxia apoptosis of transplanted BMSCs in the osteonecrotic area limits the transplantation efficacy of BMSCs[12,13]. Therefore, exploring the intervention targets to inhibit the hypoxia-induced apoptosis of BMSCs is key to improving transplantation efficacy. In this study, we confirmed that LncAABR07053481 can effectively inhibit hypoxia-induced apoptosis of BMSCs and improve the survival rate of BMSC transplantation both in vitro and in vivo. In view of this, we further evaluated the effect of BMSCs overexpressing

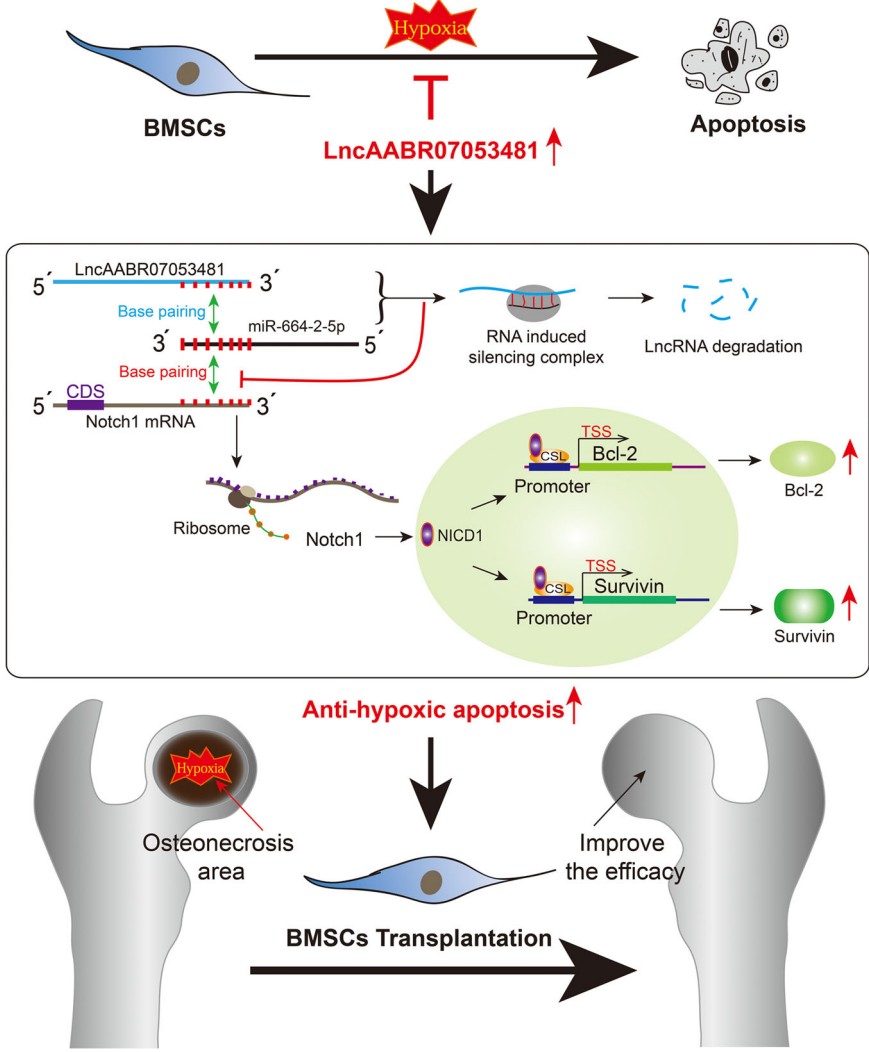

**Fig. 7 Effect and mechanism of LncAABR07053481 on the repair of early SANFH by regulating the *Notch1* pathway to inhibit hypoxia-induced apoptosis of BMSCs: LncAABR07053481 acts as a sponge for miR-664-2-5p to attenuate its repressive effect on *Notch1* and activates the *Notch1* pathway and releases NICD1 (the intracellular active fragment of *Notch1*) from the cell membrane.** NICD1 localizes to the nucleus, where it forms a transcriptionally active complex with the DNA-binding protein CSL and the coactivator Mastermind (MAM) to upregulate transcription of Notch target genes (Survivin and Bcl-2). Following which, hypoxia-induced apoptosis of BMSCs in the area of osteonecrosis is inhibited, improving the transplantation effect of BMSCs on early SANFH.

LncAABR07053481 on early SANFH in rats. The results confirmed that LncAABR07053481 overexpression significantly inhibited the hypoxia-induced apoptosis of BMSCs after transplantation and significantly improved the repair efficacy of BMSCs on early SANFH. Therefore, our results show that the LncAABR07053481 intervention strategy may represent a novel therapeutic target to effectively improve the efficacy of BMSCs in the repair of early SANFH.

In summary, we found that LncAABR07053481, a significantly downregulated LncRNA under hypoxia, can inhibit hypoxia-induced apoptosis of BMSCs. LncAABR07053481 acts as a sponge for miR-664-2-5p to attenuate its repressive effect on *Notch1* and activates the *Notch1* pathway to inhibit hypoxia-induced apoptosis of BMSCs by promoting the expression of anti-apoptotic genes. We also systematically evaluated the therapeutic effect of LncAABR07053481-overexpressing BMSC transplantation on early SANFH, which provides an effective molecular strategy and a novel target for improving the transplantation efficacy of BMSCs (Fig. 7).

## Methods

**Animals**. All the animals used in this study were provided by the Laboratory Animal Center of Guizhou Medical University (Grant No.2100733) and approved by the Experimental Animal Bioethics Committee of Guizhou Medical University, Guiyang, China. We extracted BMSCs from young Sprague-Dawley (SD) rats ($n = 18$; body weight, 20–30 g), regardless of sex, and adult male SD rats ($n = 228$; body weight, 500–600 g) were used to construct SANFH models. All procedures were carried out in strict accordance with the guidelines for the Care and Use of Experimental Animals issued by the National Institutes of Health (NIH publication No. 85-23, 1996 revised).

**Cell culture**. The bilateral tibias and femurs of SD rats ($n = 18$; body weight, 20–30 g) were obtained under anesthesia. The bone marrow cavity was washed with complete L-glutamine Dulbecco's Modified Eagle Medium (L-DMEM) (Gibco, USA) containing 10% fetal bovine serum (Gibco), 100 U/mL of penicillin, and 100 µg/mL of streptomycin (Hyclone, USA) to obtain the mixture containing bone marrow. The mixed solution was transferred to a 15-mL tube and centrifuged at 1000 rpm for 5 min. The supernatant was discarded to obtain single-cell precipitation, which was resuspended with 5 mL of complete L-DMEM, and then transferred to a 25-cm$^2$ cell culture bottle. The cells were cultured at 37 °C and 5% $CO_2$. When the cell confluence reached 90%, the cells were digested with 1 mL of 0.25% trypsin–0.02% ethylene-diaminetetraacetic acid (EDTA) (Gibco) at 37 °C and passaged at a ratio of 1:3.

**Differentiation identification of BMSCs**. Third-generation BMSCs were inoculated into six-well plates. When the cell fusion degree reached 60,100, and 60% according to the instructions of the BMSC osteogenesis induction kit (Cyagen Biosciences, USA), the BMSC adipogenic induction kit (Cyagen Biosciences) and the BMSC chondrogenic induction kit (Cyagen Biosciences), respectively, BMSC differentiation was induced, while the control group continued to be cultured with complete L-DMEM. The BMSCs were identified by staining at 2, 3, and 4 weeks after differentiation. ALP staining was used to identify osteogenic differentiation, oil red O staining was used to identify adipogenic differentiation, and Alixin blue staining was used to identify chondrogenic differentiation.

**Surface antigen identification of BMSCs**. Third-generation BMSCs were obtained, and the cell concentration was adjusted to $2.0 \times 10^7$ cells/mL. The 50 μL cell suspension was fed into the flow tube, and 5 μL surface antigen antibodies (Anti-CD29/AF647, Anti-CD90/PE-CyTM7, Anti-CD73/PE, Anti-CD45/FITC, and Anti-CD11b/V450) (BD, USA) were added in order. Next, 45 μL buffer (Hyclon) was added to each tube and incubated at room temperature for 30 min. Following incubation, the cells were washed twice with phosphate-buffered saline (PBS) (Hyclon), and 500 μL of buffer was added to each tube for detection by flow cytometry (Beckman, USA).

**Hypoxia cell model**. The proportion of mixed gas in the cell incubator (Thermo, USA) was adjusted to an oxygen concentration of 0%, nitrogen concentration of 95%, and carbon dioxide concentration of 5%. Then, BMSCs were placed in the mixed gas for continuous treatment for 48 h to induce hypoxia.

**GeneChip microarray assay**. Total RNA in cell samples from both groups was extracted with Trizol reagent (Invitrogen, USA). The microarray and data collection were conducted by KangChen Biotech (Shanghai, China), as mentioned previously[55]. The Agilent Feature Extraction software program (version 11.0.1.1; Agilent Technologies, USA) was used to analyze acquired array images. Quantile normalization and subsequent data processing were performed using the GeneSpring GX v12.1 software package (Agilent Technologies). After quantile normalization of the raw data, LncRNAs and messenger RNAs (mRNAs), among which ≥3 out of nine samples had flags in the normoxia or hypoxia group, were chosen for further data analysis. A paired t-test was employed to assess the differentially expressed LncRNAs and mRNAs between the experimental and control groups. Differentially expressed LncRNAs and mRNAs between the two groups were identified through fold change filtering (fold change >2.0). Differentially expressed LncRNAs and mRNAs with statistical significance between the two groups were identified through P value filtering ($P < 0.05$). Data visualization of LncRNA and mRNA expression levels was achieved via a heatmap using the pheatmap R package.

**Bioinformatics analysis**. MicroRNA (miRNA) binding sites on LncRNAs or mRNAs were predicted using the web-based program RNAhydrid (https://bibiserv. cebitec.uni-bielefeld.de/rnahybrid/). The Gene Set Enrichment Analysis (GSEA) version 4.1.0 tool (Broad Institute, USA; http://www.gsea-msigdb.org/gsea/index. jsp) was used for pathway analysis of gene-expression data to find enriched Kyoto Encyclopedia of Genes and Genomes (KEGG) pathways. GSEA signatures were assessed using the Molecular Signatures Database version 6.2, and all processing parameters used were the defaults adopted in GSEA version 4.1.0.

**LncRNA coding capacity prediction**. The Coding-Potential Assessment Tool (CPAT) (http://lilab.research.bcm.edu/cpat/) was used to assess the coding capacity of LncAABR07053481. The CPAT conducted a logistic regression model using the sequence features of open reading frame coverage, open reading frame size, hexamer usage bias, and Fickett TESTCODE statistic. The CPAT selected 0.364 as a cutoff for the coding probability (CP); a CP of <0.364 suggests a non-coding sequence, whereas that of ≥0.364 indicates a coding sequence[56].

**In vitro translation assay**. To validate that LncAABR07053481 is indeed a non-coding RNA, we performed in vitro translation experiments. Briefly, the LncAABR07053481 transcript was cloned into pcDNA4-myc-His-plasmid to obtain the recombinant, and pcDNA4/LncAABR07053481-myc-His-plasmid was transfected into BMSCs using eukaryotic transduction kit (Thermo) for 48 h. Meanwhile, pcDNA4/KLF4-myc-His was used as a coding protein control. The expression of Myc-fused protein was analyzed by immunoblotting with an anti-Myc antibody (Abcam ab32072, 1:1000).

**Lentiviral (LV) vector production and cell transfection**. The LV vector that knocked down or overexpressed LncAABR07053481 was purchased from Genechem (Shanghai, China). Briefly, the LV (sh AABR07053481 or Oe AABR07053481) constructs were generated based on different regions of the Rattus LncAABR07053481 sequence (NCBI accession XR_005502377.1). The same LV vector containing an insert of nonspecific RNA oligonucleotide (sh-NC or Oe-NC) was used as a negative control. Lentiviruses containing these plasmids were generated by transfecting the 293 T packaging cell line with the shRNA vector (GU6-

LncAABR07053481-GFP-puro) and overexpression vector (Ubi-LncAABR07053481-GFP-puro). Cell transfection: third-generation BMSCs were divided into groups according to transfection conditions. LV vectors (MOI = 90) were added into the BMSCs culture medium. After 12 h, the culture medium was changed to complete L-DMEM. Three days following the infection, the expression of the green fluorescent protein was observed using an inverted fluorescence microscope, and the transfection efficiency was calculated. Five days after transfection, the cells were continuously selected with a complete medium containing 2 μg/mL of puromycin (Solarbio, Beijing, China), and the efficiency of target gene knockdown or overexpression was detected by quantitative real-time polymerase chain reaction (qPCR).

**Reverse transcription and qPCR**. The total RNA was isolated from bone tissue or cells with the TRIzol reagent. Tissue samples: bone forceps crush frozen bone tissue and place it in a mortar, then add liquid nitrogen and grind the pieces into a powder. RNA was isolated from tissue powder frozen on liquid nitrogen by homogenization in Trizol Reagent. Cell samples: RNA was isolated by direct lysis of cells with Trizol Reagent. RNA extraction was performed using an RNAeasy RNA extraction kit (Sangon Biotech, Shanghai). RNA was quantified using a Nanodrop instrument. cDNAs were synthesized by PrimeScriptTM RT reagent kit (Sangon Biotech). qPCR was performed using the SYBR Premix Ex Taq II kit (Sangon Biotech) and the Applied Biosystems 7500 Fluorescent Quantitative PCR system (Applied Biosystems Life Technologies, USA). GAPDH was used to normalize the expression levels of LncAABR07053481, Notch1, and miR-664-2-5p. The transcript levels were analyzed by the $2^{-\Delta\Delta Ct}$ method. Primers (Sangon Biotech) were as follows: LncAABR07053481-F: CATGGAAGGGCTTGCTCCAT; LncAABR07053481-R: CCAGCTGAACCACCTCGAAT; Notch1-F: TGAAG-GAACGAGCCTGGGT; Notch1-R: CATTCAGGCAGGTCCCACTT; miR-664-2-5p-RT: CTCAACTGGTGTCGTGGAGTCGGCAATTCAGTTGAGCCCAATC; miR-664-2-5p-F:TCGGCAGGCTGGCTGGGGAAAA; miR-664-2-5p-R:CGTGGAGTCGGCAATTCAGTTGA.

**Western blot analysis**. Cellular protein was extracted with 1 × cell lysis buffer (Beyotime, China). Equal amounts of proteins obtained from different types of cell lysates were separated by 10 or 15% SDS-PAGE (Beyotime), transferred to PVDF membranes, and subjected to western blotting using an ECL chemiluminescence reagent (Merck Millipore). Antibodies for P53 (Abcam ab131442, 1:1000), Bcl-2 (Abcam ab196495, 1:1000), Bid (Abcam ab272880, 1:2000), Cleaved-CASP-3 (Abcam ab13847, 1:500), Survivin (Abcam ab134170, 1:1000), myc (Abcam ab32072, 1:1000), β-actin (Abcam ab8227, 1:3000), and IgG (Abcam ab6795, 1:2000) were purchased from Abcam (Cambridge, USA). Antibodies for Notch1 (Cell Signaling Technology 3608, 1:1000) and NICD1 (Cell Signaling Technology 4147, 1:1000) were purchased from Cell Signaling Technology (Boston, USA).

**RNA fluorescence in situ hybridization (RNA-FISH)**. Cells were seeded on coverslips, fixed with 4% paraformaldehyde (Solarbio) for 10 min at room temperature, and permeabilized in 70% ethyl alcohol (Solarbio) at 4 °C at least 1 h. Hybridization was performed using the LncRNA Fish Probe Mix (Ribobio, China) in a moist chamber at 37 °C for 12–16 h. As positive controls, 18S and U6 (Ribobio) were used for the cytoplasm and nucleus, respectively. Image sections were acquired using laser confocal microscopy (Leica SP5, Heidelberg, Germany).

**Luciferase assay**. As mentioned previously, after BMSCs were transfected with lentiviruses containing the luciferase reporter gene and lysed with cell lysate according to the instructions of the luciferase assay kit (Solarbio), the renilla luciferase reaction substrate was added. Finally, the luciferase activity was measured using a Dual-Luciferase reporter assay system (Promega, USA).

**RNA immunoprecipitation (RIP) assay**. A Magna RIP RNA-Binding Protein Immunoprecipitation Kit (Millipore, USA) was used to determine the relationship between LncAABR07053481 or Notch1 mRNA and miR-664-2-5p. Briefly, the cells were harvested, lysed, and then reacted with RIP buffer containing magnetic beads conjugated with anti-Argonaute2 (AGO2) and negative control immunoglobulin G (IgG) (Millipore). Following the recovery of the antibody using protein A/G beads, the coprecipitated RNAs were used for complementary DNA synthesis and evaluated by qPCR.

**ROS determination**. Cells grown in confocal Petri dishes were treated according to the conditions of each group. A DHE (Sigma, USA) probe was then added to the cells and incubated for an additional 30 min. Finally, the generation of intracellular ROS was assessed by fluorescence analysis using laser confocal microscopy.

**Mitochondrial membrane potential detection**. According to the instructions of the JC-1 mitochondrial membrane potential detection kit (KeyGen Biotech, China), the treated cells were incubated with the dye mixture at 37 °C for 30 min and then washed with PBS three times. The cells were observed under a laser confocal microscopy and recorded by a random typical visual field. The red:green fluorescence ratio was calculated by ImageJ software (version 1.4.3.67).

**ATP levels detection**. The BMSCs were collected according to the experimental groups and the instructions of the ATP Assay Kit (Beyotime). An appropriate lysate was added to the samples of each group, before centrifuging at 12,000×$g$/min for 5 min at 4 °C. The supernatant was collected as the sample to be tested. The standard and test solutions were prepared according to the instructions of the kit. Then, 20 μL test or standard samples were added successively to each well of a 96-well plate in triplicate. Next, 100 μL of the detection-working solution was added, mixed well, and incubated for 25–30 min at 37 °C or room temperature according to the kit instructions. Finally, the absorbance at 532 nm was detected by an enzyme-labeling instrument (Biotech, USA). The standard curve was drawn according to the absorbance value, and the concentration or activity of the sample was calculated using the standard curve.

**Flow cytometry assay**. Cells were cultured in a 6-well culture plate, then subjected for 48 h to hypoxia or normoxia conditions when the cells grew to 85%. The cells were washed with phosphate-buffered saline, and 5 μL of Annexin V-fluorescein isothiocyanate (FITC) and 5 μL of propidium iodide (PI) were directly added according to the instructions of the Annexin V-FITC apoptosis detection kit (BD). The cells were gently vortexed and incubated at room temperature in the dark for 15 min, then detected by flow cytometry.

**TUNNEL/DAPI detection of apoptosis**. The treated cells were digested and centrifuged, and the cell density was adjusted to $2.5 \times 10^4$/mL before their inoculation in a confocal Petri dish. The cells of each group were treated strictly in accordance with the instructions of the TUNEL (Beyotime) and DAPI (Solarbio) detection kits. The samples were observed under a confocal microscope, and images were collected.

**SANFH animal model**. SANFH models were established as previously described in ref. [13]. Briefly, adult male SD rats ($n = 228$; body weight, 500–600 g) were used to induce steroid-associated osteonecrosis with sequential injections of lipopolysaccharides and methylprednisolone. The rats were intravenously injected twice with lipopolysaccharides (2 mg/kg; Sigma) (the injection time was >30 min) once a day. From day 2, after the second injection of lipopolysaccharides, methylprednisolone (60 mg/kg; Pfizer, USA) was injected into the gluteal muscle of rats for 7 continuous days. Rats in the control group were injected with saline. Eight weeks after modeling, magnetic resonance imaging (MRI) was used to evaluate the necrosis of the femoral head area, and the successful models were used for in vivo experiments.

**Construction and transplantation of tissue-engineered bone**. The xenogeneic antigen-extracted cancellous bone (XACB) (Yapeng Biological Technology, Shanghai, China) was made into a small cylinder with a diameter of 2 mm and a height of 2 mm, before soaking in a complete L-DMEM medium for 24 h. After that, 500 μL of BMSC suspension ($1 \times 10^7$ cells/mL) was dripped into XACB and cultured in a 37 °C and 5% $CO_2$ incubator for 3 h. Following incubation, 3–4 mL of complete L-DMEM was added to Petri dishes containing XACB before proceeding with the culture and changing the medium once every 2–3 days. On the fifth day of culture, the growth of BMSCs on the surface of XACB was observed by scanning electron microscopy (Hitachi, Japan). In the same way, BMSCs overexpressing or with knockdown of LncAABR07053481 were co-cultured with XACB to construct tissue-engineered bone. One side of the femoral head of the anesthetized rat model was exposed under sterile conditions. At the junction of bone and cartilage, a sterile spherical drill with a diameter of 2 mm was used to grind from the posterolateral to the anterior medial side at a depth of ~3 mm. Then, the necrotic bone tissue was scraped off, and the constructed tissue-engineered bone was implanted into the bone defect area. There were ten rats in each group; the normal rats ($n = 28$), rats with simple necrotic bone curettage ($n = 40$), and rats implanted with tissue-engineered bone with empty vector ($n = 40$) were used as controls.

**Oxygen concentration detection**. The oxygen concentration was measured using a needle-type oxygen electrode (Hansatech Instruments, Norfolk, UK) and analyzed with the OxygraphPlus software program (Hansatech Instruments), as previously described in ref. [57]. Briefly, the oxygen electrode was calibrated at 37 °C with 0 and 100% air saturation (i.e., 20.9% oxygen of all dissolved gas by volume). Measurements were performed according to the manufacturer's instructions. Oxygen measurements were made by penetrating the bone tissue with the oxygen electrode tip. Upon introduction into the tissue, the micro-electrodes responded with a time constant that was estimated to be of the order of 10 s. A stable reading was obtained within 30 s, and upon reaching the current plateau value, the electrode was extracted from the bone tissue and the tip was maintained within the suffusing saline solution.

**Live imaging of animals**. Before BMSC transplantation, we labeled live-BMSCs with DiR fluorescent dye (YEASEN, Shanghai, China) and co-cultured them with XACB. At 48 h after BMSC transplantation, the rats were anesthetized by an intraperitoneal injection of chloral hydrate (10%, 3 mL/kg) (Leagene, Beijing, China); then, non-invasive in vivo multispectral fluorescence imaging was performed using the Spectrum in vivo imaging system (IVIS) (PerkinElmer, USA) with the Living Image software (version 4.4).

**Hematoxylin and eosin (H&E) and Masson staining**. Whole intact femurs were dissected from rats, fixed in phosphate-buffered paraformaldehyde (Solarbio) for 48 h, then decalcified for 2 weeks in 10% EDTA (Solarbio). Subsequently, the samples of each group were dehydrated with ethanol (Solarbio) solutions, adopting gradient concentrations of 50, 70, 80, 90, 95, and 100%. Finally, the femur tissue samples were embedded in paraffin and cut into 4-μm-thick sections using an ultra-thin microtome (Leica, Germany). The sections were stained according to the instructions of the H&E staining kit (Solarbio) and the Masson staining kit (Solarbio). Following staining, the sections were observed under the microscope and imaged.

**Micro-computed tomography (CT) analysis**. The femurs were dissected from rats and fixed with 4% paraformaldehyde for 24 h, before being scanned and analyzed by high-resolution micro-CT (Skyscan 1172, Germany). We used the NRecon image reconstruction software version 1.6, CTAn data analysis software version 1.9, and CTVol 3D model visualization software version 2.0 to analyze the parameters of the trabecular bone in the transplantation area. The scanner was set at 50 kVp and 201 μA, with a resolution of 6 μm. The region of interest was analyzed to determine the trabecular number (Tb.N), trabecular thickness (Tb.Th), new bone volume fraction (BV/TV), trabecular number/trabecular separation, and thickness per tissue volume.

**Immunofluorescence (IF) assay**. After treatment, femoral head tissue was fixed in 4% formaldehyde, dehydrated by ethanol gradient (Solarbio), decalcified by dec-alcified solution (Solarbio), embedded in paraffin, and finally cut into 4-μm-thick sections. Then, the tissues were incubated with anti-Runx2 or anti-OPN overnight at 4 °C. The next day, the cells were incubated with secondary antibodies conjugated to FITC and Cy3 for 30 min at 37 °C. For nuclear staining, the cells were incubated with DAPI for 10 min. Fluorescence was visualized using laser confocal microscopy.

**Statistics and reproducibility**. All statistical analyses were conducted using SPSS version 20.0 (IBM Corporation, USA) or GraphPad Prism version 7 (GraphPad Software, USA). Each experiment was performed at least in triplicate. Parametric or non-parametric statistical tests were selected by determining first if the data were normally distributed using the Kolmogorov–Smirnov normality test. If the data followed a normal distribution and their variance was uniform, then the data were presented as mean ± standard deviation values (SD), and statistical significance was determined using two-tailed unpaired Student's $t$-tests for comparison between two groups or one-way ANOVA followed by Tukey's post hoc test for multiple comparisons (≥3 groups). If the data did not follow a normal distribution pattern, they were described by M(P25 and P75) values, and the Kruskal–Wallis rank-sum test was used for comparisons between groups. If the difference between groups was statistically significant, the Dwass–Steel–Critchlow–Fligner method was further used for multiple comparisons. $P < 0.05$ was considered to be statistically significant.

**Reporting summary**. Further information on research design is available in the Nature Portfolio Reporting Summary linked to this article.

## Data availability
All raw microarray data generated in this study were deposited in NCBI's GEO under the accession number GSE225572. The numerical source data for graphs are available in Supplementary Data 1. Uncropped and unedited blots are available in Supplementary Information (Supplementary Fig. 3). Other data are available from the corresponding authors on reasonable request.

## Code availability
No custom codes/software were generated in this study. All the data analysis in this study were performed using open-source software tools.

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

## Acknowledgements

This work was supported by the National Natural Science Foundation of China (Grant No. 82060397, 82260434, and 82260429); Guizhou Provincial Natural Science Foundation (Grant No. Qiankehejichu[2020]1Y311); Postgraduate Research Fund Project of Guizhou Provincial Department of Education (Grant No. Qianjiaohe YJSCXJH[2020] 141); and Science and Technology Foundation of Guizhou Provincial Health Committee (Grant No. gzwkj2021-232, gzwkj2021-234, and gzwjkj2020-1-130). We thank the clinical medicine research center of The Affiliated Hospital of Guizhou Medical University for providing the experimental site and equipment. Thank the teachers of this center for their guidance. Additionally, we thank LetPub (www.letpub.com) for its linguistic assistance during the preparation of this manuscript.

## Competing interests

The authors declare no competing interests.
