## [Peer Review File · Communications Biology]

Reviewers' comments:

Reviewer #2 (Remarks to the Author):

In this manuscript, the authors reported that a novel lncRNA LncAABR07053481, could inhibit hypoxia-induced apoptosis of BMSCs by functioning as a competing endogenous RNA for miR-664-2-5p that silences Notch1 mRNA. Their work provides an anti-apoptotic target to improve the efficacy of BMSC transplantation in early SANFH. Overall, the topic of this manuscript is interesting, but the presentation is not well clear or organized. There are several issues that need to be improved:

1. This manuscript needs to be paid particular attention to language grammar and sentence structure so that the goals and results of the study are easy for the readers. The title describing the core of the article is not concise. Several sentences contain grammatical mistakes or are not complete sentences. For example, "a major problem in clinical treatment" in line 23, "At week 12" in line 68, "a large number of hypoxia-induced apoptosis of BMSCs" in line 322, "inhibits apoptosis" in line 411, and so on.

2. In general, there is a lack of detailed explanation of essential methods used in the study. The descriptions should be added for SANFH animal model, oxygen concentration detection, tissue RNA expression assays, hypoxia culture, flow cytometry assay, and so on. Please provide exact complete and standardized information for antibodies, lentivirus vectors, and other vital reagents. The methods of several bioinformatic analyses in this study should clearly be described, including statistical test methods and vital parameters.

3. The terminology of the manuscript needs careful editing by someone with expertise in technical English editing, especially in the biology or medical research field. Inaccurate descriptions should be avoided, such as "double antibody" in line 100, "RNA sequencing" in line 76, "bioinformatics tool software" in line 136, "small animal live imaging" in line 381, and so on.

4. BMSCs, as a type of primary cells, are not immortalized cells with homogeneous, consistent traits. In the section of lentivirus infection, how to construct BMSCs to be cell lines with stable knockdown or overexpression of the gene? Please provide a basis or evidence to support that lentivirus stably knockdown or overexpress the gene in BMSCs for two weeks or longer time.

5. Appropriate comparisons between groups in statistical tests should be considered to support their corresponding conclusions. For example, comparisons of the difference between the "Sg AABR07053481" group and the "Vec" group should be clearly shown in Figure 2A.

6. In the manuscript, are the images chosen or displayed appropriately? For example, the fluorescence images in Figures 2L and 6F, and the graphs including Figure 3B-C and Figure 4B should be clearer. The color tones of Figure S1A, C-E seem abnormal. Please explain it. The generation of the "control" data in Figure S1B should be explained in the legend.

7. In this manuscript, the four quadrants in flow cytometry don't add up to 100%. Whether it was an error when analyzing the data? According to the results, the apoptosis rate of group "Oe AABR07053481 Hypoxia" basically returned to normal levels. However, the expression of apoptosis-related proteins in this group is far from the normal group. How can this be explained?

Reviewer #3 (Remarks to the Author):

Steroid-associated osteonecrosis of the femoral head (SAONFH) is a complicated condition in the musculoskeletal system.

The mechanism of the onset and development of SAONFH is unclear so far.

Certain attempts have been made to treat SAONFH at the early stage.

In this study, Zhang et al. found that LncAABR07053481 could promote the repair process of the early-stage SAONFH via regulating the Notch1 signaling pathway.

These findings are novel.

However, some concerns are came up.

1. The title says LncAABR07053481 promotes repair of early steroid-induced avascular necrosis of the

femoral head and SD rats were used in the animal experiment. However, I can not find the protocol where the SAONFH rat models were established throughout the manuscript.

2. Line 255, the necrotic bone tissue was scraped off.

How did the author determine which part of the femoral head is necrotic? With what tools? Micro-CT or Micro-MRI?

The diameter of the rat femoral head is approximate 4 mm in Figure 6A. The diameter of the drill is 2.0 mm in Line 254.

How could the necrotic bone be scraped off in such a narrow space? Is there any edge of the removed necrotic bone in micro-CT images or pathological sections?

3. There are some incorrect expressions in the manuscript.

What is the double antibody in Line 100? Did the author mean penicillin-streptomycin?

Please have the manuscript carefully checked and well polished with the help of the English native speakers or academic English language editing companies.

Responses to Reviewer:

Reviewer #2 (Remarks to the Author):

In this manuscript, the authors reported that a novel lncRNA LncAABR07053481, could inhibit hypoxia-induced apoptosis of BMSCs by functioning as a competing endogenous RNA for miR-664-2-5p that silences Notch1 mRNA. Their work provides an anti-apoptotic target to improve the efficacy of BMSC transplantation in early SANFH. Overall, the topic of this manuscript is interesting, but the presentation is not well clear or organized. There are several issues that need to be improved:

1. This manuscript needs to be paid particular attention to language grammar and sentence structure so that the goals and results of the study are easy for the readers. The title describing the core of the article is not concise. Several sentences contain grammatical mistakes or are not complete sentences. For example, “a major problem in clinical treatment” in line 23, “At week 12” in line 68, “a large number of hypoxia-induced apoptosis of BMSCs” in line 322, “inhibits apoptosis” in line 411, and so on.

Responses: We thank you for your time spent reviewing and revising this manuscript and for your valuable comments. We would like to apologize for the grammatical and sentence structure errors in the original manuscript. We have revised the manuscript using scientific language according to your suggestions, and the precedent version of the title has been replaced with “*LncAABR07053481 promotes the repair of steroid-induced avascular necrosis of the femoral head by inhibiting hypoxia-induced apoptosis of BMSCs*”. We also corrected grammatical mistakes or

incomplete sentences. In addition, the manuscript has been professionally reviewed and edited for proper English language, grammar, punctuation, spelling, and overall style by highly qualified native English-speaking editors of LetPub (www.letpub.com). If you need certificates, we can upload the relevant supporting materials as supplementary files.

2. In general, there is a lack of detailed explanation of essential methods used in the study. The descriptions should be added for SANFH animal model, oxygen concentration detection, tissue RNA expression assays, hypoxia culture, flow cytometry assay, and so on. Please provide exact complete and standardized information for antibodies, lentivirus vectors, and other vital reagents. The methods of several bioinformatic analyses in this study should clearly be described, including statistical test methods and vital parameters.

Responses: We have revised the manuscript as you suggested. We have supplemented and described in detail the basic experimental methods used in the study, such as the SANFH animal model, oxygen concentration detection, tissue RNA expression assays, hypoxia culture, flow cytometry assay, and bioinformatic analyses (Page 7, line 121-125; Page 8, line 135-150; Page 9, line 167-172; Page 10, line 173-177; Page 13, line 245-248; Page 14, line 249-251; Page 14, line 258-268; Page 16, line 288-299, and so on). We have also perfected the information provided on antibodies, lentiviral vectors, and other vital reagents. For the statistical analysis, we have normatively rewritten the methods of the Statistical analysis section (Page 7-8, line 126-141; Page

9, line 158-161; Page 11, line 194-200; Pages 18-19, line 336-349).

3. The terminology of the manuscript needs careful editing by someone with expertise in technical English editing, especially in the biology or medical research field. Inaccurate descriptions should be avoided, such as “double antibody” in line 100, “RNA sequencing” in line 76, “bioinformatics tool software” in line 136, “small animal live imaging” in line 381, and so on.

Responses: We have revised some inaccurate or erroneous descriptions in the whole manuscript (Page 4, line 72-73; Page 5, line 94-95; Page 7-8, line 126-141; Page 16-17, line 300-306, and so on). We have corrected similar mistakes one by one under the guidance of native English-speaking editors (The changes in the revised manuscript have been highlighted in red).

4. BMSCs, as a type of primary cells, are not immortalized cells with homogeneous, consistent traits. In the section of lentivirus infection, how to construct BMSCs to be cell lines with stable knockdown or overexpression of the gene? Please provide a basis or evidence to support that lentivirus stably knockdown or overexpress the gene in BMSCs for two weeks or longer time.

Responses: Yes, as you said, BMSCs, as a type of primary cells, are not immortalized cells with homogeneous, consistent traits. Therefore, the BMSCs used in this study were indeed not cell lines. Generally, the third-generation BMSCs were transfected with overexpression- or knockdown lentivirus; then, the BMSCs successfully

transfected with lentivirus were screened by puromycin, and qPCR was used to verify that the target gene was overexpressed or knocked down in BMSCs. Finally, BMSCs successfully transfected with lentivirus were cryopreserved or used in experiments; at the same time, BMSCs successfully transfected with lentivirus continued to be subcultured, and puromycin was added to continuously screen the cells. When the cells were cultured to the seventh passage (>2 weeks), we confirmed by qPCR that the expression of the target gene was still upregulated >10 times in the overexpressed BMSCs, while it was still significantly down-regulated in the knocked down BMSCs (see figure below). Therefore, knockdown or overexpression of the target gene in BMSCs successfully infected with lentivirus can be maintained for >2 weeks, but whether it can be maintained for longer is uncertain because cells beyond the eighth passage were not used in the experiment.

Successfully transfected cells				analyze				
groups		GAPDH-CT	AABR07053481-CT	Mean GAPDH-CT	ΔCt	Mean ΔCt	$\Delta \Delta Ct$	$2^{-\Delta \Delta Ct}$
control	1	14.78255562	22.59727775	14.5889233	8.008354	8.0042	0.004176	0.997109286
	2	14.71312601	22.73766289		8.14874		0.144562	0.904654237
	3	14.59062471	22.34665975		7.757736		-0.24644	1.186277508
	4	14.26938688	22.69080477		8.101881		0.097703	0.934519399
Vec	1	14.54545732	22.23673208	14.5883511	7.648381		-0.3558	1.279692329
	2	14.79543443	22.78807947		8.199728		0.19555	0.873239683
	3	14.4562212	22.5892819		8.000931		-0.00325	1.002253301
	4	14.55629139	22.51674551		7.928394		-0.07578	1.053933297
Oe	1	14.38593707	18.69661855	14.5587863	4.137832		-3.86635	14.58431492
	2	14.47975097	18.62668538		4.067899		-3.93628	15.30868947
	3	14.93792535	18.72097842		4.162192		-3.84199	14.34012644
	4	14.43153172	18.61139561		4.052609		-3.95157	15.4717947
Sg	1	14.27900021	30.01499939	14.3121433	15.70286		7.698678	0.004813566
	2	14.1711478	29.71899986		15.40686		7.402679	0.005909785
	3	14.68904691	29.4640007		15.15186		7.147679	0.007052354
	4	14.10937834	29.73500061		15.42286		7.418679	0.005844603

Third or fourth generation cells after transfection				analyze				
groups		GAPDH-CT	AABR07053481-CT	Mean GAPDH-CT	ΔCt	Mean ΔCt	$\Delta \Delta Ct$	$2^{-\Delta \Delta Ct}$
control	1	13.8798822	23.54394971	14.0871548	9.456795	9.3097	0.147072	0.903081596
	2	14.06526383	23.28031251		9.193158		-0.11657	1.084150803
	3	14.56504715	23.47344584		9.386291		0.076568	0.948310958
	4	13.83842586	23.28980377		9.202649		-0.10707	1.077041762
Vec	1	14.54966582	23.57588427	14.4705161	9.105368		-0.20435	1.152171112
	2	14.30550554	23.82097842		9.350462		0.040739	0.97215674
	3	14.48810389	23.47032075		8.999805		-0.30992	1.239637682
	4	14.53878903	23.59138768		9.120872		-0.18885	1.139856009
Oe	1	14.23585009	20.0133375	14.3615925	5.651745		-3.65798	12.62295826
	2	14.40775475	20.0984405		5.736848		-3.57288	11.89988024
	3	14.36944792	20.37421186		6.012619		-3.2971	9.829402986
	4	14.43331706	19.87937864		5.517786		-3.79194	13.85118027
Sg	1	13.68325755	30.74382153	14.1731803	16.57064		7.260918	0.006519974
	2	14.41086764	31.32567522		17.15249		7.842772	0.004356026
	3	14.21123997	30.22070986		16.04753		6.737806	0.009369538
	4	14.38735618	30.8522327		16.67904		7.36932	0.006048027

5. Appropriate comparisons between groups in statistical tests should be considered to support their corresponding conclusions. For example, comparisons of the difference between the “Sg AABR07053481” group and the “Vec” group should be clearly shown in Figure 2A.

Responses: Thank you for pointing this out. We have corrected some of the graphs based on the data (figure 2A).

6. In the manuscript, are the images chosen or displayed appropriately? For example, the fluorescence images in Figures 2L and 6F, and the graphs including Figure 3B-C and Figure 4B should be clearer. The color tones of Figure S1A, C-E seem abnormal. Please explain it. The generation of the “control” data in Figure S1B should be explained in the legend.

Responses: When we uploaded the manuscript, we compressed the figures, so the resolution of some figures was reduced. We have re-uploaded clearer figures (figures 2L and 6F; figures 3B-C; figure 4B). For the abnormal color tone in Figure S1A, to make the image more beautiful and the cell morphology resolution clearer, we adjusted the tone parameters of the microscope and displayed the image with a green background after acquisition. If you feel that this might be inappropriate, we can replace the current image with one that has a normal white balance background. In Figure S1C, the white balance tone was used when we collected images, so the display background is gray. As for Figures S1D, E, all images were observed under white light. To identify the multidirectional differentiation potential of BMSCs,

BMSCs were stained with oil red O and Alixin blue after adipogenesis induction and chondrogenesis induction, respectively, and orange–red lipid droplets and blue acid mucopolysaccharides could be observed under an inverted microscope. Based on your suggestion, we have annotated the control group in the Figure 1B legend (Page 47, line 911-912).

7. In this manuscript, the four quadrants in flow cytometry don't add up to 100%. Whether it was an error when analyzing the data? According to the results, the apoptosis rate of group “Oe AABR07053481 Hypoxia” basically returned to normal levels. However, the expression of apoptosis-related proteins in this group is far from the normal group. How can this be explained?

Responses: Since the data in the flow cytometry graph were automatically retained with three significant digits, they are not exact values, so the four quadrants in flow cytometry do not add up to 100%. Our experimental data were repeated ≥ 3 times, so a great number of flow analysis images were generated. For the overexpression group, when we selected the flow images, we chose the images that could most intuitively show the statistical differences. Now, we have replaced the flow images and selected more representative images (figure 2D).

Responses to Reviewer:

Reviewer #3 (Remarks to the Author):

Steroid-associated osteonecrosis of the femoral head (SAONFH) is a complicated condition in the musculoskeletal system. The mechanism of the onset and development of SAONFH is unclear so far. Certain attempts have been made to treat SAONFH at the early stage. In this study, Zhang et al. found that LncAABR07053481 could promote the repair process of the early-stage SAONFH via regulating the Notch1 signaling pathway. These findings are novel. However, some concerns are came up.

1. The title says LncAABR07053481 promotes repair of early steroid-induced avascular necrosis of the femoral head and SD rats were used in the animal experiment. However, I can not find the protocol where the SAONFH rat models were established throughout the manuscript.

Responses: We thank you for your time spent reviewing and revising this manuscript and for your valuable comments. We have supplemented the detailed protocol of establishing SANFH rat models in the revised manuscript, and we have supplemented and described in detail the basic experimental methods used in the study (Page 7, line 121-125; Page 8, line 135-150; Page 9, line 167-172; Page 10, line 173-177; Page 13, line 245-248; Page 14, line 249-251; Page 14, line 258-268; Page 16, line 288-299, and so on).

2. Line 255, the necrotic bone tissue was scraped off. How did the author determine

which part of the femoral head is necrotic? With what tools? Micro-CT or Micro-MRI? The diameter of the rat femoral head is approximate 4 mm in Figure 6A. The diameter of the drill is 2.0 mm in Line 254. How could the necrotic bone be scraped off in such a narrow space? Is there any edge of the removed necrotic bone in micro-CT images or pathological sections?

Responses: All the rat SANFH models were numbered by the random number table method, and the femoral head necrosis sites of rats were located by MRI. The hip joint was exposed and the joint capsule was dissected to expose the femoral head and neck (See Figure 1 below). At the junction of the femoral head and femoral neck, drilling was performed according to the results of the preoperative MRI (See Figure 2 below). After drilling, we used a scraping spoon with a diameter of 1.8–2 mm to clean the necrotic bone at the drilling site (See Figure 3 below). For the border or the remaining necrotic bone tissue that was not cleaned, we continued to use a low-speed grinding drill with a diameter of 1–2 mm to further clean the surrounding of the necrotic area until the bone slag that was cleaned out was normal bone tissue. Indeed, due to the limitations of surgical instruments and equipment, it was difficult for us to conduct real-time localization during the operation, but we did our best to accurately locate the osteonecrotic area and clean up the necrotic bone tissue as much as possible. In addition, micro-CT was used to scan the operative area again after the operation to confirm that the necrotic bone had been cleaned; otherwise, the samples were not included in the test. Finally, when we collected micro-CT images or pathological sections, we showed the location of the resectioned edge or repaired edge of the

necrotic bone.

3. There are some incorrect expressions in the manuscript. What is the double antibody in Line 100? Did the author mean penicillin-streptomycin? Please have the manuscript carefully checked and well polished with the help of the English native speakers or academic English language editing companies.

Responses: We would like to apologize for the incorrect expressions in the original

manuscript. We have revised the manuscript using scientific language according to your suggestions (Page 4, line 72-73; Page 5, line 94-95; Page 7-8, line 126-141; Page 16-17, line 300-306, and so on). In addition, the manuscript has been professionally reviewed and edited for proper English language, grammar, punctuation, spelling, and overall style by highly qualified native English-speaking editors of LetPub (www.letpub.com). If you need certificates, we can upload the relevant supporting materials as supplementary files.

Reviewers' comments:

Reviewer #2 (Remarks to the Author):

In this manuscript, the language has been improved, and the results of the study are displayed better. However, there are still some major concerns.

1. As the authors reported, the noncoding capacity of lncRNAs was defined as a CP of <0.364 in the result by CPAT (described in line 156-157). However, the data in Figure 1C showed that the sequence ABR07053481 has a CP of >0.4 . Please explain this discrepancy with careful proof.

2. It has been known that CRISPR-Cas9 genome editing makes alterations at the genomic level by using a target specific crRNA hybridized to the tracrRNA (that is the gRNA), which is complexed to the Cas9 protein. Therefore, in the comparisons of Figure 2A and other graphs, the "Vec" group should not be used as the same control group in the different experiments, i.e., lncRNA-overexpressing lentivirus transfection and CRISPR-cas9 genome editing. Please provide more information about the transfections.

3. There was a lack of methodological description about the in vitro translation assay in Figure 1D.

4. In line 134, did the groups have 9 samples? In line 137, "the two samples" should be "the two group".

5. The numbers in Venn plots of Figure 4B should be labeled.

Reviewer #3 (Remarks to the Author):

Zhang et al explored the function of lncAABR07053481 in the repair of steroid-induced avascular necrosis of the femoral head. This study is of novelty and the manuscript is well organized. And the author made enough revisions according to my previous comment.

Here are some minor suggestions.

First of all, the onset and development of steroid-induced avascular necrosis of the femoral head is a dynamic process. So is the repair process. Hence, samples from a single time point are usually considered insufficient to have a better understanding of the repair effect. I suggest the author have a dynamic evaluation of the repair process at different time points in the following study.

Secondly, as we all know, not all SD rats will suffer steroid-induced avascular necrosis of the femoral head after the GC administration. In addition, the administration of large dose GC is fatal to some rats in certain cases. Hence, I suggest the authors declare the survival rates of GC-treated rats.

Moreover, the success rate of steroid-induced avascular necrosis of the femoral head model is highly recommended to declare in the manuscript as well.

Reviewers' comments:

Reviewer #2 (Remarks to the Author):

In this manuscript, the language has been improved, and the results of the study are displayed better. However, there are still some major concerns.

1. As the authors reported, the noncoding capacity of lncRNAs was defined as a CP of <0.364 in the result by CPAT (described in line 156-157). However, the data in Figure 1C showed that the sequence ABR07053481 has a CP of >0.4 . Please explain this discrepancy with careful proof.

Responses: We thank you for taking the time to review our manuscript. According to the reference (reference 28), the noncoding capacity of lncRNAs was defined as $CP < 0.364$ in the CPAT result. However, when we performed CPAT to predict the coding ability of lnc ABR07053481, the result showed that $CP = 0.43278906836867$, and the result of coding label was "NO" (see the following figures). In addition, we also predicted the protein coding potential of lnc ABR07053481 based on the Coding Potential Calculator (CPC) online tool, and the prediction results also suggested "noncoding" (see the following figures). The actual prediction results were different from the theoretical results, so we further determined the coding ability of lnc ABR07053481 by in vitro translation assay. Finally, we identified lnc ABR07053481 as a non-coding RNA based on online prediction results and in vitro translation experiments.

Results of CPAT predictions:

Result for species name : mm10 with job ID : 1620717051							
Data ID	Sequence Name	RNA Size	ORF Size	Fickett Score	Hexamer Score	Coding Probability	Coding Label
0	AABR07053481	2372	483	0.8502	-0.00831230733358	0.43278906836867	no

Result for species name : mm10 with job ID : 1620719063							
Data ID	Sequence Name	RNA Size	ORF Size	Ficket Score	Hexamer Score	Coding Probability	Coding Label
0	LNC	10490	333	0.789	0.00779562838589	0.17127263726202	no

Result for species name : mm10 with job ID : 1620717425							
Data ID	Sequence Name	RNA Size	ORF Size	Ficket Score	Hexamer Score	Coding Probability	Coding Label
0	β-ACTIN	4211	387	1.2244	0.463430433919	0.81170081254625	yes

Result for species name : mm10 with job ID : 1620716383							
Data ID	Sequence Name	RNA Size	ORF Size	Ficket Score	Hexamer Score	Coding Probability	Coding Label
0	BAX-202	522	522	1.2331	0.426010786546	0.91650217848748	yes

Results of CPC predictions:

ID	C/NC	CODING POTENTIAL SCORE	EVIDENCE	UTR-DB HITS	RNA-DB HITS
AABR07053481	noncoding (weak)	-0.410205	detail	search	search

ID	C/NC	CODING POTENTIAL SCORE	EVIDENCE	UTR-DB HITS	RNA-DB HITS
Lnc	noncoding (weak)	-0.136386	detail	search	search

ID	C/NC	CODING POTENTIAL SCORE	EVIDENCE	UTR-DB HITS	RNA-DB HITS
Bax-202	coding	4.49183	detail	1	0

2. It has been known that CRISPR-Cas9 genome editing makes alterations at the genomic level by using a target specific crRNA hybridized to the tracrRNA (that is the gRNA), which is complexed to the Cas9 protein. Therefore, in the comparisons of Figure 2A, the “Vec” group should not be used as the same control group in the different experiments, i.e., lncRNA-overexpressing lentivirus transfection and CRISPR-cas9 genome editing. Please provide more information about the transfections.

Responses: Thank you for pointing out this issue. We have realized that this is a mistake, and we are sorry that we mistakenly wrote "Sg" instead of "sh". We have revised the issue you pointed out in the new revised manuscript (Figure 2A-M; Figure 3H-N; Figure 4F-G; Figure 6A-H; Page 9-10, line 167-178; Page 22, line 413-415; Page 22, line 420; Page 23, line 428; Page 23, line 439; Page 29, line 526; Page 30, line 568; Page 30, line 570; Page 30, line 578; Page 30, line 581; , and so on), and we

have added detailed information about lentivirus transfection in the materials and methods (Page 9-10, line 167-178). As you mentioned, the CRISPR-Cas9 genome editing system uses Single-guide RNA (that is the SgRNA) to identify target gene sequences and guides Cas9 protein to perform targeted cutting of DNA, thus achieving the purpose of knockout gene expression at the DNA level. We found that after checking the product records of lentivirus construction company and our experimental records, we used shRNA technology to knock down the target gene, mainly because the basic expression of the target gene CT value was stable at about 20. After consulting the company of lentivirus construction, it was confirmed that shRNA technology could be performed. Therefore, we constructed the shRNA lentivirus by using the same series of plasmid vectors as the overexpressed lentivirus. The principle of shRNA technology is indeed different from that of SgRNA. The shRNA combines with RNA-induced silencing complex (RISC), recognizes RNA possession of complementary sequences, and leads to RNA degradation. The shRNA interferes at the post-transcription level of genes and plays a knock-down role, and shRNA can be integrated into the genome through viral vectors, and can be stably expressed and play a role for a long time. We have fully revised this in the manuscript.

3. There was a lack of methodological description about the in vitro translation assay in Figure1D.

Responses: In the Materials and Methods section, the in vitro translation assay

method is supplemented in detail (Page 9, line 159-166).

4. In line 134, did the groups have 9 samples? In line 137, “the two samples” should be “the two group”.

Responses: We thank you for pointing out this issue, we’ve changed “the two samples” to “the two group” (Page 8, line 138).

5. The numbers in Venn plots of Figure 4B should be labeled.

Responses: According to your suggestion, we have modified the Venn plots of Figure 4B. All comments have been corrected. We will be happy to edit the text further based on helpful comments from you.

Reviewer #3 (Remarks to the Author):

Zhang et al explored the function of LncAABR07053481 in the repair of steroid-induced avascular necrosis of the femoral head. This study is of novelty and the manuscript is well organized. And the author made enough revisions according to my previous comment.

Here are some minor suggestions.

First of all, the onset and development of steroid-induced avascular necrosis of the femoral head is a dynamic process. So is the repair process. Hence, samples from a single time point are usually considered insufficient to have a better understanding of the repair effect. I suggest the author have a dynamic evaluation of the repair process at different time points in the following study.

Responses: We thank you for taking the time to review our manuscript. Yes, it is very necessary to dynamically evaluate the repair process of osteonecrosis. Thank you very much for your advice, which is of great reference value for our future research, we will dynamically evaluate the repair process of osteonecrosis at multiple different time points in the following studies, and we will gradually make up for the deficiencies in the future research.

Secondly, as we all know, not all SD rats will suffer steroid-induced avascular necrosis of the femoral head after the GC administration. In addition, the administration of large dose GC is fatal to some rats in certain cases. Hence, I suggest the authors declare the survival rates of GC-treated rats. Moreover, the

success rate of steroid-induced avascular necrosis of the femoral head model is highly recommended to declare in the manuscript as well.

Responses: According to your suggestion, the data on the survival rate of SD rats treated with GC and the success rate of steroid-induced avascular necrosis of the femoral head model have been described in the supplementary data. The success rate of using lipopolysaccharide (LPS) combined with glucocorticoid to construct a rat model of femoral head necrosis was between 60% and 70%. As for the death of SD rats, our previous results showed that SD rats were continuously injected with GC (60mg/Kg) for 10 to 14 days, which did not cause death in SD rats. The death of SD rats was caused by LPS injection, and the mortality rate was about 30%. The death of SD rats mainly occurred within two days after the injection of LPS (2mg/Kg). After two days, with the continuous use of GC, the SD rats generally did not die. Therefore, when we used LPS combined with GC to construct the SANFH model, the cause of rat death was mainly due to septic shock caused by LPS. However, when the GC injection was used alone, almost no deaths occurred in SD rats. If death occurred, the mortality rate was generally less than 10%. We will be happy to edit the text further based on helpful comments from you.

REVIEWERS' COMMENTS:

Reviewer #2 (Remarks to the Author):

No further comments

Reviewer #3 (Remarks to the Author):

Zhang et al explored the function of LncAABR07053481 in the repair of steroid-induced avascular necrosis of the femoral head. This study is of novelty and the manuscript is well organized. And the author made enough revisions according to my previous comments. I think it could be published in the present version in COMMSBIO.